# Global burden and future trends of inguinal, femoral, and abdominal hernia in older adults: A systematic analysis from the Global Burden of Disease Study 2021

Jinwei Zhang[1,2], Jiaxuan Wang[3], Xiangyu Han[4], Junjie Fan[4], Chao Huang[4], Yonghong Dong [1,2,4]*

1 The Gastrointestinal, Pancreatic, Hernia and Abdominal Wall Surgery Department of, Shanxi Provincial People's Hospital, Shanxi Medical University, Taiyuan, China, 2 The Fifth Clinical Medical College, Shanxi Medical University, Taiyuan, China, 3 The First Clinical Medical College, Changzhi Medical College, Changzhi, China, 4 Shanxi Provincial People's Hospital, Taiyuan, China

* youthdong007@163.com

## Abstract

### Objective

This study aims to comprehensively evaluate the global, regional, and national burden, trends, and health inequalities of inguinal, femoral, and abdominal hernias among older adults from 1990 to 2021, conduct predictive analyses, and provide insights to inform future public health strategies.

### Methods

A secondary analysis was conducted using the Global Burden of Disease 2021, focusing on the temporal trends, health inequality, and predictive development of inguinal, femoral, and abdominal hernia burden among older adults.

### Results

Globally, the number of incident cases of inguinal, femoral, and abdominal hernias among older adults continuously increased from 1990 to 2021, along with the decline in age-standardized rates, prevalence, and Disability-Adjusted Life Years (DALYs). Older males exhibited higher incidence rates, prevalence, and DALYs for hernias relative to females. In terms of the Socio-Demographic Index (SDI) from 1990 to 2021, the Age-Standardized Prevalence Rate (ASPR) and Age-Standardized Rate of Disability-Adjusted Life Years (ASDR) remained the highest in low-middle and low SDI regions, while the Age-Standardized Incidence Rate (ASIR) was the highest in high SDI regions. At the national level, 10 countries experienced a significant increase in ASDR and ASPR, and 15 countries in ASIR. Among these, the highest increase was observed for ASIR in China, ASPR in Georgia, and ASDR in American

**Data availability statement:** All relevant data are within the article and its supporting information files. This study is based on the Global Burden of Disease (GBD) database, and all relevant data and associated files are also publicly available through the corresponding official website (https://www.healthdata.org/research-analysis/gbd).

**Funding:** The author(s) received no specific funding for this work.

**Competing interests:** The authors have declared that no competing interests exist.

**Abbreviations:** ASIR: Age-Standardized Incidence Rate; ASPR: Age-Standardized Prevalence Rate; ASDR: Age-Standardized Disability-Adjusted Life Year Rate; DALYs: Disability-Adjusted Life Years; AAPC: Average Annual Percentage Change; SDI: Socio-Demographic Index; UI: Uncertainty Interval; ICD: International Classification of Diseases; APC: Annual Percentage Change; BAPC: Bayesian Age-Period-Cohort; INLA: Integrated Nested Laplace Approximation; SII: Slope Index of Inequality; TFU25: Total Fertility Rate Under 25; EDU15+: Mean Education Level for Those Aged 15 and Above; LDI: Lag-Distributed Income

Samoa. The projections to the year 2035 indicate an increase in the incidence and prevalence of hernias, with older males remaining predominant. However, the DALY rate is expected a declining trend.

## Conclusions

In spite of the progress in reducing the burden of inguinal, femoral, and abdominal hernias in older adults, the overall burden tends to rise. In particular, countries such as India, China, and Georgia are experiencing an increasing burden. It is crucial to implement targeted medical interventions, especially for older males in these regions.

## Introduction

The inguinal, femoral, and abdominal hernia poses severe global health challenges, with a prevalence of up to 32.5 million cases by 2019 [1], and an especially significant impact on the population aged over 60 [2]. Demographically, as the global population ages the burden of hernia tends to weigh more in this age group, making up a larger part of the overall burden of diseases [3,4].

Recent advancements in the diagnosis, surgery, and perioperative management of inguinal, femoral, and abdominal hernia have contributed to reduction in both morbidity and mortality [3,5,6]. However, there remains a lack of research on the burden of hernia among older adults on global and regional scale. The burden of hernia varies significantly across regions with different SDI levels [7,8]. High SDI regions typically benefit from better healthcare infrastructure and access to surgical treatment and correspondingly lower morbidity and mortality [9]. In contrast, the insufficient medical resource in low and lower-middle SDI regions exacerbates the burden of the disease, especially among older people. There is presently an urgent need for macroscale research on the burden of hernia among older adults globally and across various SDI regions in order to develop effective interventions tailored to the specific needs of each region. Such studies are crucial in understanding the disease's impacts, deciding public health interventions, and guiding clinical practice.

The Global Burden of Disease Study (GBD) 2021 provides new perspectives for understanding disease distribution, trends, and risk factors [10,11]. To the best of our knowledge, the comprehensive analysis of the burden of hernia among older adults based on the latest GBD 2021 data has not yet been conducted. In this background, our study aims to comprehensively evaluate the global, regional, and national burden of inguinal, femoral, and abdominal hernias among older adults from 1990 to 2021, utilizing data from the GBD 2021.

The analysis focuses on trends in incidence, prevalence, and DALYs across 204 countries and territories, stratified by geographic regions, SDI levels, age groups, and gender. By identifying and quantifying disparities in the disease burden, particularly across different SDI regions, our study seeks to provide actionable insights into the unequal distribution of healthcare resources and outcomes. The findings aim to inform the development of tailored public health policies and strategies that prioritize

the needs of older adults, especially in low- and lower-middle SDI regions where the burden is disproportionately high. Additionally, the results are intended to serve as a baseline for policymakers, healthcare providers, and global health organizations to design evidence-based interventions, allocate resources effectively, and address health disparities in aging populations. The ultimate goal is to implement a series of effective measures to alleviate the burden of hernias among older adults, thereby reducing the strain on societal healthcare systems.

## Methods

### Data sources

Our study leveraged data from the GBD 2021 and utilized DisMod-MR 2.1 to synthesize inguinal, femoral, and abdominal hernia. This model employs differential equations to enhance the accuracy of findings [12]. Inguinal, femoral, and abdominal hernia is classified in the International Classification of Diseases (ICD) 10 by codes K40-K42.9, K44-K46.9, and in the ICD 9 by codes 550–551.1, 551.3–552.1, 552.3–553.1, 553.3–553.9 [1]. In this study, the term 'hernia' encompasses the types delineated in the ICD-10 and ICD-9 classifications. Based on the World Health Organization's "World Report on Ageing and Health" and prior research on the disease burden among older adults using the GBD framework, our study defines "older adults" as individuals aged 60 years and older [13,14]. This definition is consistent with global public health standards and widely recognized in studies examining ageing and its associated health impacts.

Our study extracted the estimates and 95% uncertainty interval (UI) for prevalence, incidence, and DALYs related to inguinal, femoral, and abdominal hernias. This dataset from the GBD 2021 encompasses global levels, five SDI-level regions, 21 GBD regions, and 204 countries and territories from 1990 to 2021.

### Statistics analysis

**Age-standardized rate.** To ensure comparability across regions and populations, age-standardized indicators-including the ASIR, ASPR, and ASDR- were calculated using the World Health Organization standard population. This age standardization process adjusts for differences in population age structures, enabling meaningful temporal and geographical comparisons. The calculation methodology involves applying age-specific rates to the WHO standard population distribution, with demographic data derived from the GBD 2021 dataset. Participants were categorized into eight age groups (60–64, 65–69, 70–74, 75–79, 80–84, 85–89, 90–94, and 95–99 years) for analyzing the distribution of inguinal, femoral, and ventral hernias. Age-Standardized proportions were calculated by determining the ratio of cases within each age stratum relative to the corresponding standardized population cohort [14]. This stratification allows systematic examination of hernia epidemiology across distinct geriatric age segments while maintaining demographic comparability through standardized weighting procedures. ASRs were reported per 100,000 population, with data presented as values with 95% UI.

**Joinpoint regression.** The joinpoint regression model was used to analyze the ASIR, ASPR, and ASDR temporal trends for the burden of inguinal, femoral, and abdominal hernia among older adults from 1990 to 2021. Joinpoint regression analysis partitions the continuous time-series data into multiple linear segments, each representing a statistically distinct trend over time. Within each segment, a separate linear model is fitted to estimate the temporal trend, and the number and location of joinpoints are determined using a Monte Carlo permutation test. This test assesses whether adding a joinpoint significantly improves model fit [15]. For each segment, the model calculates the annual percent change (APC) and its 95% confidence interval to quantify the direction and magnitude of the trend. The average annual percent change (AAPC) is then computed to summarize the overall trend across the entire study period [16].

**Socio-demographic index.** The SDI is an aggregate measure formulated by GBD researchers to evaluate the socio-economic conditions impacting health indicators across different regions [17]. It is calculated as the geometric mean of indices for the total fertility rate under 25 (TFU25), mean education level for those aged 15 and above (EDU15+), and lag-distributed income (LDI) per capita [11]. The SDI values in the GBD 2021 study were scaled from 0 to 1. This composite

indicator reflects a region's socio-economic health and progress, where higher SDI values indicate better socio-economic conditions and improved health outcomes. Regions are divided into 5 quintiles according to the 2021 SDI: low (0–0.466), low-middle (0.466–0.619), middle (0.619–0.712), high-middle (0.712–0 810), and high (0.810–1) [18]. These quintiles were utilized to explore disparities in disease burden across varying socio-demographic development levels. The ASIR, ASPR, and ASDR were employed in correlation analyses to investigate the relationship between the SDI and the burden of hernias among the elderly population. The analysis encompassed 204 countries and territories, as well as 21 regions. The strength and direction of the relationship between SDI and the burden of hernias among older adults were assessed by calculating Pearson correlation coefficients. A positive coefficient value indicates a direct relationship, whereas a negative value signifies an inverse relationship. This methodological approach enables a comprehensive evaluation of the association between socio-demographic development levels and hernia-related health outcomes in aging populations.

**Health inequality.** The ASRs and numbers of prevalence, incidence, and DALYs of inguinal, femoral, and abdominal hernia among older adults were extracted for inequality analysis. To quantify and characterize the socioeconomic inequality in the burden of elderly hernias across countries, we employed two complementary indices: the Slope Index of Inequality (SII) and the Concentration Index (CI). These indices were selected due to their methodological robustness and their ability to capture both the magnitude and direction of inequality across the full socioeconomic spectrum.

The SII is an absolute measure of inequality derived from a weighted linear regression model, where countries are ranked by their SDI and weighted by population size. It estimates the absolute difference in the health burden between the hypothetical lowest and highest ends of the socioeconomic hierarchy. By incorporating data from all countries rather than only comparing extreme groups, the SII provides a comprehensive assessment of the socioeconomic gradient in health outcomes. The higher absolute SII values indicate greater inequality [19]. The CI was calculated by fitting a Lorenz concentration curve to the observed cumulative relative distribution of the populations ranked by SDI and the prevalence of disease, as well as numerically integrating the area under the curve. A negative index indicates that the burden is higher in low-income countries. Conversely, a positive index indicates a higher burden in high-income countries [20]. The Lorenz concentration curve is used to assess socioeconomic-related inequality in health outcomes. The X-axis represents the cumulative population ranked by a socioeconomic indicator, while the Y-axis shows the cumulative proportion of the health variable. The 45-degree line indicates perfect equality. A curve below the line suggests the burden is concentrated among lower-SDI populations; above indicates concentration among higher-SDI groups. The degree of deviation from the equality line reflects the magnitude of inequality. This curve is often used alongside the concentration index to quantify and interpret health disparities across socioeconomic gradients [21].

**Bayesian age period cohort model.** We employed a Bayesian age-period-cohort (BAPC) model to analyze the temporal trends of hernia among older adults [22]. This model is based on the age-period-cohort model, which makes predictions based on the assumption that the effects of age, period, cohort are approximated at adjacent time points in the same study population and within the same study period. To smooth the prior estimates of age, period, and cohort effects, we specified second-order random walk (RW2) priors, which allow for flexible yet structured temporal evolution [18]. The BAPC model was implemented using the Integrated Nested Laplace Approximation (INLA) method to approximate the marginal posterior distributions. INLA enables efficient and accurate Bayesian inference for latent Gaussian models, and effectively circumvents the convergence and mixing issues commonly encountered in traditional Bayesian approaches based on Markov Chain Monte Carlo (MCMC) techniques. Compared to MCMC, INLA typically provides reliable posterior estimates with substantially reduced computational burden [23]. By incorporating both historical data and prior distributions, the BAPC model with INLA allows for more accurate estimation and projection of hernia incidence in older populations while explicitly accounting for age, period, and cohort effects.

**Statistical software and analysis.** Temporal trends were assessed using joinpoint software (version 5.0.2) from the National Cancer Institute. All statistical analyses and mapping were performed using R statistical software (version 4.3.3). A two-sided P value < 0.05 was set as the significance threshold.

### Ethical statement

This study is based on publicly available data from the GBD 2021 study, which is coordinated by the Institute for Health Metrics and Evaluation (IHME). All data used in this analysis are aggregated, de-identified, and available in the public domain (https://ghdx.healthdata.org/gbd-2021), this study meets the criteria for exemption from full ethical review according to the guidelines provided by Shanxi Provincial People's Hospital Ethics Committee and the University of Washington's IRB, which has oversight over the GBD project. The study was conducted in accordance with the Declaration of Helsinki and relevant guidelines and regulations.

## Results

### Global burden trends

The numbers and rates of inguinal, femoral, and abdominal hernias among older adults are presented in Table 1. In 2021, on global scale, the number of incident cases, prevalent cases, and DALYs among older adults due to hernias were 1.48 million, 3.62 million, and 0.811 million. The ASIR, ASPR, and ADSR among older adults were 134.87, 330.86, and 76.23 cases per 100,000 population worldwide. Compared to 1990, these rates had significantly decreased, but the number of cases had markedly increased. This trend is generally similar among older males and older females, but older males bear relatively heavier burden than older females (Fig 1 and Table 1).

Between 1990 and 2021, the global burden of hernias among older adults showed an overall decline in the ASIR, ASPR, and ADSR. The ASIR decreased from 190.28 to 134.87 per 100,000 population (AAPC = -1.12, 95% CI: -1.24 to -0.99, $P < 0.001$), the ASPR declined from 516.42 to 330.86 per 100,000 (AAPC = -1.44, 95% CI: -1.48 to -1.40, $P < 0.001$), and the ASDR dropped from 127.62 to 76.23 per 100,000 (AAPC = -1.64, 95% CI: -1.77 to -1.51, $P < 0.001$). The joinpoint regression analyses also revealed the significant inflection points in ASDR in 1994, 1997, 2000, 2006, and 2014. In spite of the inflection points, the ASDR displayed a general downward trend from 1990 to 2021. While the ASIR and ASPR showed an overall declining trend, the ASIR (APC = 1.87, 95% CI: 0.96 to 2.79, $P < 0.001$) and ASPR (APC = 0.49, 95% CI: 0.18 to 0.80, $P = 0.004$) demonstrated an upward trend following the inflection point in 2019. A similar trend of ASIR and ASPR was also observed among older males (APC in ASIR = 2.05, 95% CI: 0.97 to 3.14, $P = 0.001$; APC in ASPR = 0.33, 95% CI: -0.04 to 0.71, $P = 0.077$) and older females (APC in ASIR = 1.16, 95% CI: 0.78 to 1.53, $P < 0.001$; APC in ASPR = 1.33, 95% CI: 0.94 to 1.72, $P < 0.001$) (Fig 2 and Table 1).

### Burden trends in association with the SDI

The only high SDI region saw significant reductions in ASDR, ASIR and ASPR (AAPC in ASIR = -1.51, 95% CI: -1.74 to -1.27, $P < 0.001$; AAPC in ASPR = -2.00, 95% CI: -2.10 to -1.90, $P < 0.001$; AAPC in ASDR = -2.28, 95% CI: -2.55 to -2.01, $P < 0.001$) from 1990 to 2021. When pooled together, all five SDI regions exhibited a significant downward trend. Notably, the low-middle SDI and low SDI regions showed the highest ASPR and ASDR in 1990, while the high SDI region had the highest ASIR. This pattern reappeared in 2021. Additionally, the middle SDI region had the highest cases of DALY and prevalence in 2021, indicating the greatest burden of hernia among older adults, and the highest cases of the incident were reported by the high SDI region in 1990 and 2021, reflecting the heaviest incidence burden (Fig 1 and Table 1). At the regional or national level, the significant negative correlations of SDI and ASPR (R = -0.60, $P < 0.001$) and ASDR (R = -0.59, $P < 0.001$) were observed, along with the significant positive correlation of SDI to ASIR (R = 0.27, $P < 0.001$) (Figs 3 and 4). Furthermore, there was a significant negative correlation of SDI to AAPC of ASIR (R = -0.43, $P < 0.001$), ASPR (R = -0.41, $P < 0.001$), and ASDR (R = -0.50, $P < 0.001$), indicating the association of higher SDI to greater declines in AAPC (Fig 5).

Table 1. Incidence, prevalence, and DALYs of inguinal, femoral, and abdominal hernias among older adults in 1990 and 2021, and AAPC from 1990 to 2021 on global scale.

| Location | Incidence | | | | | Prevalence | | | | | DALYs | | | | |
|---|---|---|---|---|---|---|---|---|---|---|---|---|---|---|---|
| | Incident cases (1990) | ASR of Incidence (1990) | Incident cases (2021) | ASR of Incidence (2021) | AAPC 1990–2021 | Prevalent cases (1990) | ASR of Prevalence (1990) | Prevalent cases (2021) | ASR of Prevalence (2021) | AAPC 1990–2021 | DALYs cases (1990) | ASR of DALYs (1990) | DALYs cases (2021) | ASR of DALYs (2021) | AAPC 1990–2021 |
| Global | 943797 (597192–1401851) | 190.28 (120.67 to 281.98) | 1480996 (939275–2168972) | 134.87 (85.68 to 197.11) | -1.12 (-1.24 to -0.99) | 2519472 (1852442–3287447) | 516.42 (381.70 to 671.09) | 3616357 (2610365–4812502) | 330.86 (239.41 to 439.52) | -1.44 (-1.48 to -1.40) | 578681 (481515–717124) | 127.62 (106.62 to 157.34) | 811296 (683554–970553) | 76.23 (64.26 to 91.00) | -1.64 (-1.77 to -1.51) |
| **Sex** | | | | | | | | | | | | | | | |
| Female | 138468 (87779–204503) | 161.14 (98.84 to 245.30) | 195591 (124409–287931) | 33.35 (21.21 to 49.09) | -1.44 (-1.53 to -1.35) | 585304 (438751–754451) | 219.70 (165.15 to 282.60) | 806014 (588461–1059953) | 137.58 (100.41 to 180.94) | -1.51 (-1.55 to -1.46) | 214890 (182176–263030) | 84.59 (71.53 to 103.30) | 291287 (242025–341384) | 49.46 (41.11 to 57.98) | -1.72 (-1.84 to -1.60) |
| Male | 805329 (508386–1200068) | 359.72 (227.64 to 534.19) | 1285406 (816998–1888868) | 250.95 (159.97 to 367.29) | -1.17 (-1.31 to -1.02) | 1934168 (1410100–2548513) | 891.13 (655.66 to 1166.10) | 2810343 (2014719–3765988) | 556.06 (400.93 to 741.95) | -1.52 (-1.56 to -1.48) | 363791 (287369–487963) | 185.88 (148.37 to 248.30) | 520009 (421700–649534) | 108.70 (88.57 to 135.06) | -1.73 (-1.85 to -1.60) |
| **Sociodemographic Index** | | | | | | | | | | | | | | | |
| High SDI | 381986 (239781–564747) | 263.29 (165.26 to 389.87) | 439871 (282052–641478) | 162.89 (103.86 to 238.80) | -1.51 (-1.74 to -1.27) | 587898 (428659–778036) | 405.89 (295.10 to 538.29) | 599359 (418571–827191) | 218.06 (150.59 to 303.87) | -2.00 (-2.10 to -1.90) | 138921 (120770–159326) | 96.98 (83.95 to 111.34) | 142914 (116809–168170) | 47.68 (39.02 to 56.64) | -2.28 (-2.55 to -2.01) |
| High-middle SDI | 258965 (164814–382298) | 200.05 (127.55 to 295.06) | 353531 (225376–520124) | 136.43 (87.08 to 200.37) | -1.22 (-1.30 to -1.13) | 568947 (415400–750504) | 451.11 (331.55 to 591.78) | 706007 (492874–967449) | 273.75 (191.55 to 374.62) | -1.60 (-1.63 to -1.57) | 104584 (89138–123303) | 90.59 (77.59 to 105.56) | 122760 (100354–149956) | 48.87 (40.01 to 59.50) | -1.98 (-2.06 to -1.90) |
| Middle SDI | 147608 (92017–221208) | 117.51 (73.23 to 175.77) | 376639 (240732–556570) | 111.29 (71.31 to 163.89) | -0.27 (-0.37 to -0.16) | 575198 (420336–758650) | 473.56 (348.16 to 621.59) | 1079830 (785845–1428886) | 323.49 (236.52 to 426.44) | -1.24 (-1.28 to -1.19) | 126756 (100557–165715) | 118.84 (94.61 to 155.29) | 240480 (201410–284917) | 76.45 (64.07 to 90.27) | -1.43 (-1.53 to -1.32) |
| Low-middle SDI | 116492 (71574–177239) | 161.14 (98.84 to 245.30) | 235506 (147512–351486) | 135.10 (84.69 to 201.11) | -0.57 (-0.60 to -0.53) | 580957 (428248–760702) | 827.86 (612.85 to 1080.94) | 866767 (637752–1133510) | 506.54 (374.06 to 660.48) | -1.61 (-1.67 to -1.56) | 152078 (115148–215870) | 236.01 (177.61 to 336.55) | 219746 (181184–285530) | 138.83 (114.37 to 180.32) | -1.72 (-2.01 to -1.42) |
| Low SDI | 36870 (22187–56906) | 138.07 (82.93 to 213.16) | 73799 (45955–110343) | 127.80 (79.50 to 190.88) | -0.27 (-0.37 to -0.16) | 202055 (144852–272798) | 779.92 (559.34 to 1050.80) | 360660 (258605–482555) | 637.86 (458.10 to 852.10) | -0.69 (-0.81 to -0.57) | 55392 (40569–86075) | 240.06 (173.36 to 381.25) | 84573 (62447–127871) | 166.40 (122.26 to 254.13) | -1.17 (-1.51 to -0.84) |
| **Region** | | | | | | | | | | | | | | | |
| Andean Latin America | 3975 (2525–5830) | 163.38 (103.55 to 239.84) | 8365 (5624–12049) | 115.41 (77.64 to 166.06) | -1.12 (-1.21 to -1.03) | 16477 (12339–21319) | 683.63 (512.17 to 883.94) | 28781 (21687–37132) | 397.85 (300.09 to 512.83) | -1.74 (-1.76 to -1.73) | 3379 (2515–4383) | 145.61 (108.46 to 188.68) | 5997 (4590–7749) | 84.15 (64.46 to 108.66) | -1.76 (-2.04 to -1.47) |

*(Continued)*

Table 1. (Continued)

| Location | Incidence | | | | | Prevalence | | | | | DALYs | | | | |
|---|---|---|---|---|---|---|---|---|---|---|---|---|---|---|---|
| | Incident cases (1990) | ASR of Incidence (1990) | Incident cases (2021) | ASR of Incidence (2021) | AAPC 1990–2021 | Prevalent cases (1990) | ASR of Prevalence (1990) | Prevalent cases (2021) | ASR of Prevalence (2021) | AAPC 1990–2021 | DALYs cases (1990) | ASR of DALYs (1990) | DALYs cases (2021) | ASR of DALYs (2021) | AAPC 1990–2021 |
| **Australasia** | 5266 (3457–7615) | 167.97 (110.28 to 242.97) | 8981 (5936–12917) | 127.39 (83.49 to 184.42) | -0.88 (-0.93 to -0.82) | 11538 (8251–15530) | 370.82 (265.06 to 499.24) | 16901 (12068–22987) | 236.10 (166.69 to 324.04) | -1.45 (-1.46 to -1.43) | 2555 (2108–3071) | 85.13 (70.16 to 102.13) | 4597 (3646–5569) | 60.47 (47.88 to 73.70) | -1.00 (-1.70 to -0.29) |
| **Caribbean** | 5291 (3535–7684) | 162.36 (108.48 to 235.67) | 7783 (5265–11150) | 116.17 (78.55 to 166.50) | -1.06 (-1.13 to -0.98) | 19114 (14155–24729) | 587.38 (435.22 to 759.74) | 26269 (19538–33794) | 391.83 (291.28 to 504.27) | -1.31 (-1.34 to -1.28) | 5353 (4543–6328) | 172.79 (146.55 to 204.14) | 7169 (5856–8633) | 106.10 (86.66 to 127.81) | -1.53 (-1.95 to -1.12) |
| **Central Asia** | 7603 (4993–11132) | 125.05 (81.30 to 184.12) | 10072 (6602–14818) | 94.79 (62.07 to 138.86) | -0.89 (-0.99 to -0.78) | 26553 (19871–34451) | 456.60 (342.87 to 590.28) | 33021 (23446–44245) | 321.83 (231.31 to 427.46) | -1.14 (-1.20 to -1.08) | 4628 (3849–5595) | 82.53 (68.91 to 99.35) | 4792 (3786–6002) | 50.13 (40.11 to 62.00) | -1.68 (-2.17 to -1.19) |
| **Central Europe** | 83505 (54094–120558) | 416.91 (269.86 to 603.51) | 61377 (40837–88063) | 204.61 (135.79 to 294.32) | -2.26 (-2.29 to -2.22) | 189842 (147236–238941) | 976.79 (758.79 to 1225.71) | 115239 (83649–153575) | 383.30 (277.26 to 512.85) | -2.98 (-3.02 to -2.95) | 36769 (32012–42276) | 202.20 (176.59 to 230.78) | 19730 (16390–23844) | 64.72 (53.57 to 78.49) | -3.63 (-3.89 to -3.38) |
| **Central Latin America** | 32176 (20532–47219) | 325.96 (207.69 to 479.14) | 54652 (37050–76335) | 175.85 (119.34 to 245.28) | -1.97 (-2.05 to -1.90) | 116945 (91146–146716) | 1208.14 (942.62 to 1513.51) | 200399 (157370–248467) | 648.04 (509.62 to 802.22) | -1.99 (-2.05 to -1.93) | 27424 (24260–31013) | 303.79 (269.50 to 341.95) | 52769 (45594–60583) | 174.91 (151.27 to 200.47) | -1.78 (-2.06 to -1.50) |
| **Central Sub-Saharan Africa** | 2592 (1572–3929) | 101.48 (61.27 to 153.80) | 5307 (3444–7730) | 87.79 (56.58 to 127.53) | -0.55 (-0.79 to -0.31) | 12739 (9179–16987) | 519.61 (373.86 to 692.74) | 28965 (22161–37155) | 492.25 (375.51 to 630.85) | -0.24 (-0.27 to -0.21) | 3222 (2001–5553) | 153.81 (93.32 to 275.38) | 7094 (4093–11808) | 141.05 (79.34 to 244.73) | -0.29 (-0.38 to -0.20) |
| **East Asia** | 64312 (38007–100647) | 58.46 (34.65 to 91.15) | 234812 (140785–367012) | 82.05 (49.49 to 127.61) | 1.08 (1.02 to 1.15) | 222678 (145407–321591) | 208.66 (137.72 to 298.76) | 479109 (299101–715798) | 168.78 (106.16 to 250.90) | -0.71 (-0.75 to -0.66) | 25589 (18028–36322) | 28.28 (20.43 to 39.62) | 46532 (30655–68758) | 17.44 (11.70 to 25.40) | -1.61 (-1.83 to -1.38) |
| **Eastern Europe** | 74945 (46343–112668) | 197.17 (121.71 to 297.02) | 87893 (54655–133067) | 178.67 (111.06 to 270.12) | -0.33 (-0.45 to -0.21) | 184758 (132932–246430) | 501.93 (363.25 to 666.09) | 184080 (128973–252150) | 377.57 (265.66 to 515.54) | -0.91 (-1.04 to -0.79) | 33002 (28241–38995) | 93.94 (80.74 to 110.10) | 36620 (30859–43445) | 76.60 (64.70 to 90.58) | -0.66 (-1.33 to 0.02) |
| **Eastern Sub-Saharan Africa** | 9101 (5471–14122) | 103.41 (61.98 to 160.34) | 21045 (13298–31142) | 110.07 (69.31 to 162.83) | 0.12 (-0.07 to 0.32) | 50716 (36108–68711) | 578.66 (411.34 to 782.86) | 104936 (76365–139006) | 553.79 (402.55 to 732.40) | -0.13 (-0.21 to -0.05) | 13123 (8999–25741) | 172.48 (116.63 to 349.19) | 24865 (17127–47525) | 148.84 (101.47 to 291.80) | -0.47 (-0.53 to -0.40) |
| **High-income Asia Pacific** | 45173 (27691–68633) | 176.23 (108.35 to 267.24) | 98523 (60061–151524) | 174.78 (105.31 to 271.30) | -0.03 (-0.14 to 0.09) | 77988 (52877–109677) | 312.00 (213.36 to 435.88) | 131256 (86444–188713) | 222.58 (142.80 to 326.43) | -1.13 (-1.25 to -1.00) | 11456 (9157–14367) | 48.12 (38.59 to 59.78) | 20376 (14758–26509) | 28.36 (20.18 to 38.41) | -1.69 (-2.12 to -1.26) |

*(Continued)*

**Table 1.** (Continued)

| Location | Incidence | | | | | Prevalence | | | | | DALYs | | | | |
|---|---|---|---|---|---|---|---|---|---|---|---|---|---|---|---|
| | Incident cases (1990) | ASR of Incidence (1990) | Incident cases (2021) | ASR of Incidence (2021) | AAPC 1990–2021 | Prevalent cases (1990) | ASR of Prevalence (1990) | Prevalent cases (2021) | ASR of Prevalence (2021) | AAPC 1990–2021 | DALYs cases (1990) | ASR of DALYs (1990) | DALYs cases (2021) | ASR of DALYs (2021) | AAPC 1990–2021 |
| High-income North America | 61956 (38340–93656) | 131.45 (81.15 to 199.33) | 83442 (53220–124126) | 93.88 (59.70 to 139.85) | -1.08 (-1.14 to -1.02) | 109543 (76619–149787) | 231.81 (161.14 to 318.57) | 129943 (89734–182177) | 145.82 (100.37 to 204.93) | -1.46 (-1.54 to -1.37) | 37689 (33164–42110) | 79.92 (70.16 to 89.45) | 42779 (36213–48993) | 46.97 (39.89 to 53.82) | -1.61 (-1.87 to -1.36) |
| North Africa and Middle East | 28912 (17099–45351) | 137.98 (81.69 to 216.39) | 63974 (39066–97549) | 115.06 (70.67 to 174.49) | -0.57 (-0.73 to -0.41) | 63944 (39971–95182) | 311.76 (195.90 to 462.16) | 154896 (99635–224178) | 284.05 (184.61 to 407.96) | -0.32 (-0.38 to -0.26) | 8663 (6075–12381) | 46.68 (32.90 to 67.08) | 16915 (11699–24304) | 33.52 (23.56 to 47.86) | -1.06 (-1.21 to -0.92) |
| Oceania | 272 (167–411) | 77.21 (47.31 to 116.78) | 569 (365–844) | 66.70 (42.63 to 98.59) | -0.46 (-0.51 to -0.40) | 1236 (861–1675) | 370.39 (260.01 to 498.65) | 2488 (1807–3279) | 301.81 (220.24 to 395.77) | -0.65 (-0.70 to -0.60) | 143 (93–257) | 49.18 (32.32 to 90.26) | 302 (194–526) | 41.02 (26.17 to 72.96) | -0.59 (-0.69 to -0.49) |
| South Asia | 117651 (71264–178720) | 176.95 (106.95 to 269.26) | 268202 (161369–410923) | 149.36 (90.04 to 228.23) | -0.57 (-0.63 to -0.50) | 639935 (473794–833195) | 994.17 (738.92 to 1291.06) | 910114 (643762–1222204) | 517.66 (367.78 to 693.37) | -2.12 (-2.23 to -2.01) | 176418 (131752–243584) | 300.37 (222.10 to 417.96) | 220489 (170204–295726) | 135.38 (103.92 to 181.53) | -2.57 (-2.98 to -2.16) |
| Southeast Asia | 42463 (26259–64468) | 141.23 (87.16 to 214.58) | 105821 (70554–149888) | 130.35 (86.94 to 184.12) | -0.24 (-0.29 to -0.19) | 170191 (121263–227611) | 585.50 (418.16 to 781.29) | 368846 (279812–472286) | 466.05 (354.87 to 594.38) | -0.73 (-0.80 to -0.67) | 57140 (39620–89784) | 213.55 (147.42 to 336.07) | 118621 (83915–148052) | 163.55 (116.32 to 204.18) | -0.88 (-0.95 to -0.80) |
| Southern Latin America | 8093 (5340–11826) | 134.28 (88.60 to 196.12) | 14448 (9810–20269) | 129.39 (87.78 to 181.75) | -0.08 (-0.30 to 0.13) | 26986 (20331–34682) | 452.37 (341.58 to 580.34) | 41704 (31909–53316) | 372.05 (283.97 to 476.69) | -0.63 (-0.67 to -0.58) | 8039 (6969–9250) | 142.50 (123.42 to 163.61) | 9579 (8046–11300) | 84.06 (70.50 to 99.35) | -1.62 (-2.32 to -0.91) |
| Southern Sub-Saharan Africa | 2417 (1506–3608) | 74.10 (46.14 to 110.46) | 4971 (3251–7218) | 70.70 (46.28 to 102.21) | -0.19 (-0.32 to -0.05) | 12749 (9435–16655) | 395.65 (293.64 to 515.45) | 24845 (19151–31499) | 357.41 (276.51 to 451.38) | -0.33 (-0.37 to -0.28) | 3102 (2252–4479) | 104.26 (75.55 to 150.82) | 6944 (5707–8325) | 109.39 (90.05 to 130.80) | 0.15 (-0.05 to 0.36) |
| Tropical Latin America | 22532 (14180–33058) | 200.88 (126.24 to 294.97) | 45303 (29629–65202) | 138.74 (90.74 to 199.55) | -1.19 (-1.24 to -1.14) | 100617 (77898–127030) | 911.83 (706.29 to 1149.77) | 176645 (137650–221333) | 544.82 (425.17 to 681.67) | -1.64 (-1.70 to -1.58) | 17474 (14859–20446) | 170.20 (145.22 to 197.80) | 51955 (45301–58905) | 164.57 (143.40 to 186.34) | -0.09 (-0.35 to 0.17) |
| Western Europe | 313515 (196900–462142) | 412.07 (258.59 to 609.22) | 270895 (175864–388950) | 236.27 (152.00 to 342.46) | -1.73 (-2.12 to -1.35) | 403224 (295851–531198) | 525.91 (383.44 to 695.93) | 340519 (243937–459737) | 288.57 (203.13 to 395.05) | -1.93 (-1.99 to -1.87) | 89709 (77350–103478) | 116.36 (99.72 to 134.70) | 85716 (70335–100362) | 62.82 (51.43 to 74.74) | -1.92 (-2.17 to -1.66) |
| Western Sub-Saharan Africa | 12046 (7330–18479) | 112.44 (68.27 to 172.53) | 24560 (15676–36133) | 110.01 (70.10 to 161.63) | -0.17 (-0.38 to 0.05) | 61697 (43338–84078) | 593.86 (418.58 to 806.67) | 117402 (85899–155505) | 543.68 (399.29 to 717.18) | -0.29 (-0.46 to -0.13) | 13803 (9427–27934) | 148.64 (100.63 to 307.07) | 27450 (18932–53900) | 142.46 (97.72 to 286.56) | -0.14 (-0.19 to -0.09) |

Rates are reported per 100,000 person-years. Data in parentheses are 95% uncertainty intervals for cases and age-standardized rates of incident, prevalence, and DALYs, and 95% confidence intervals for AAPCs.

DALYs, disability-adjusted life years; AAPC, average annual percent change; SDI, socio-demographic index; UI, uncertainty interval.

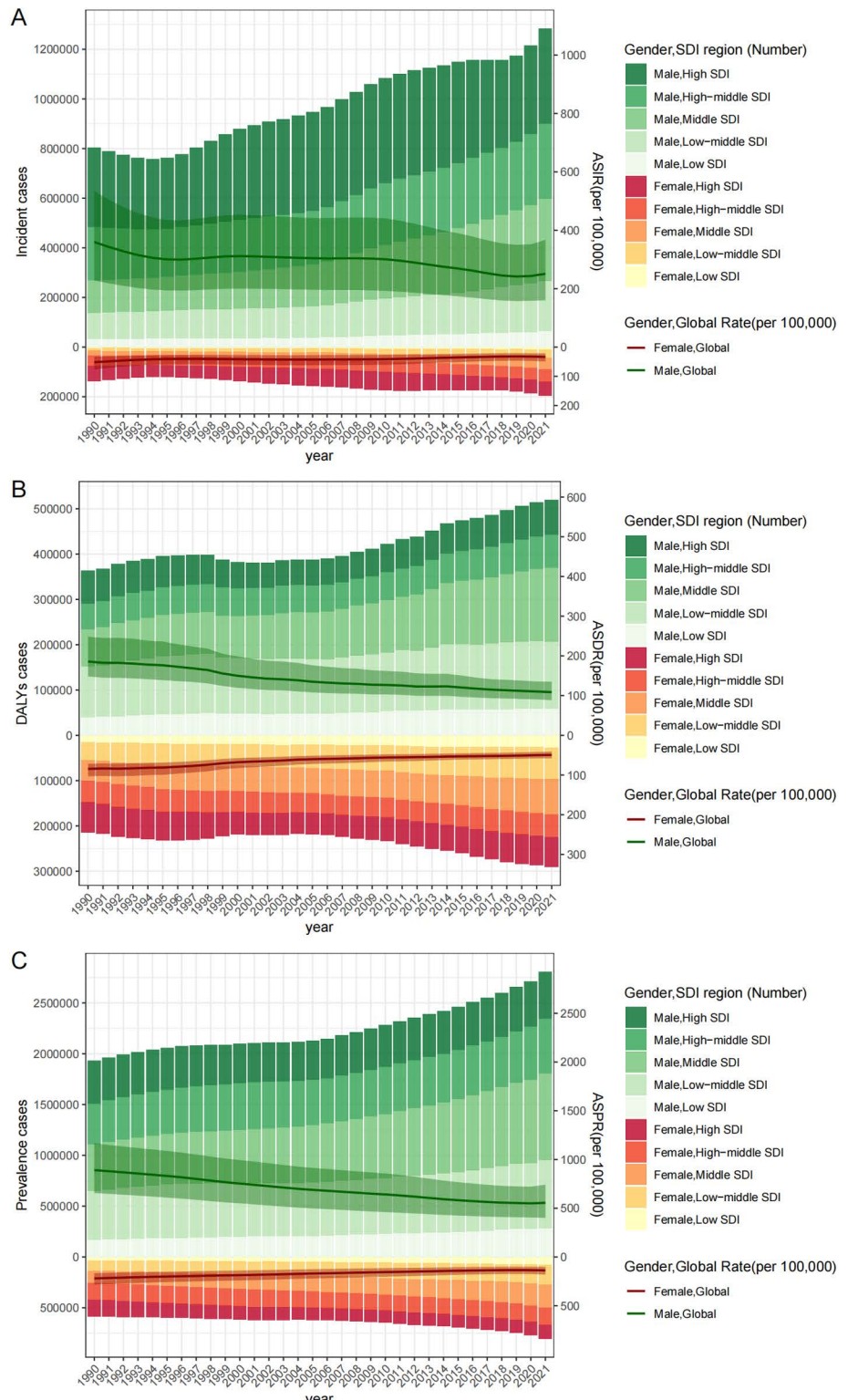

**Fig 1. Number of incident cases (A), DALYs (B), and prevalent cases (C) among older adults (by male and female, and in combination), and ASR of incidence (A), DALYs (B), and prevalence (C) related to inguinal, femoral, and abdominal hernias on global scale and at SDI level from 1990 to 2021.** This figure displays the number of incident cases, DALYs, and prevalent cases among older adults by gender, and ASR of incidence,

DALYs, and prevalence related to inguinal, femoral, and abdominal hernias on global scale and at SDI level from 1990 to 2021. DALYs: Disability-Adjusted Life Years; SDI: Sociodemographic Index; ASIR: age-standardized Incidence rate; ASDR: age-standardized disability-adjusted life years rate; ASPR: age-standardized prevalence rate.

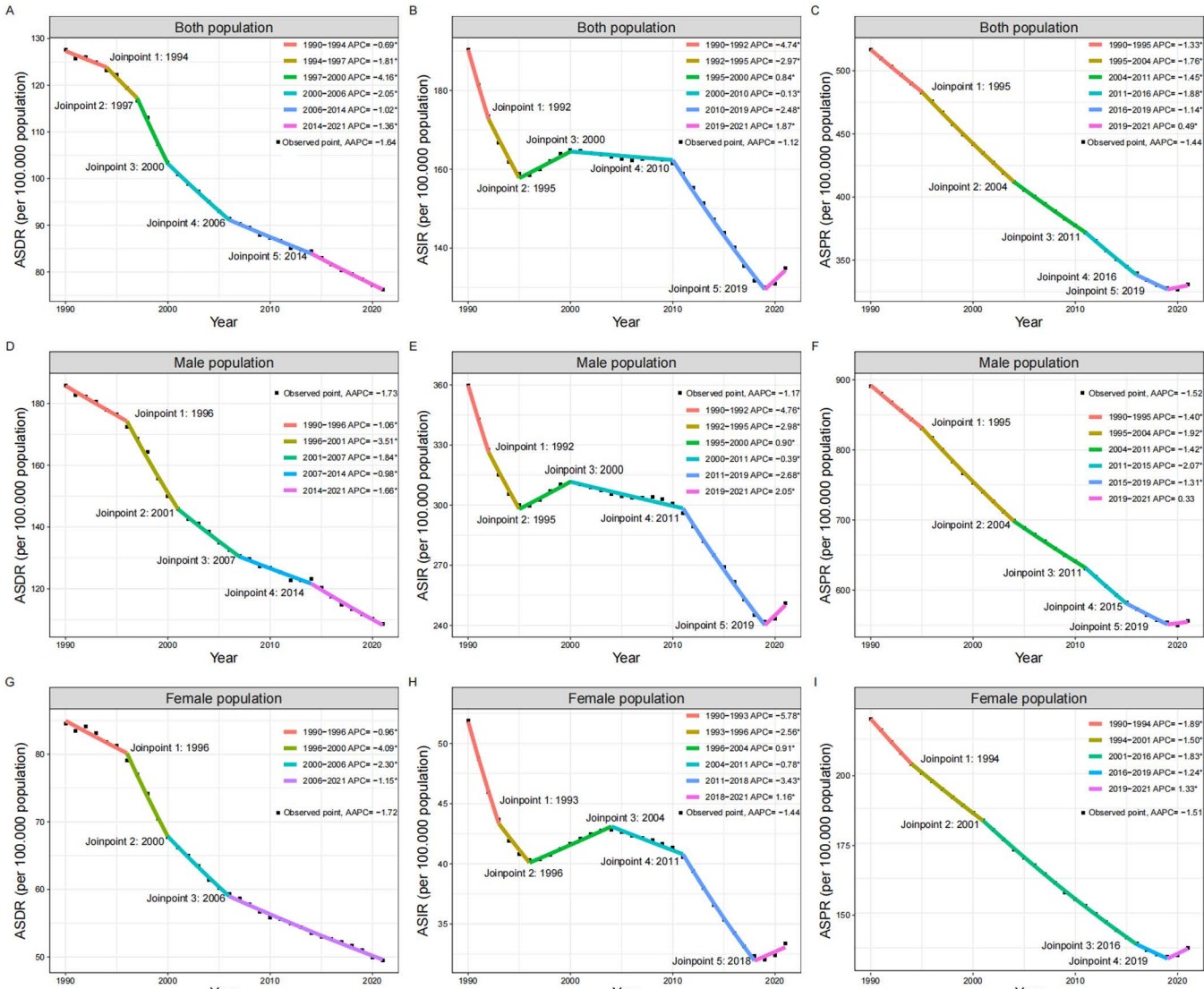

**Fig 2. Global trends in inguinal, femoral, and abdominal hernias among older adults from 1990 to 2021 using joinpoint regression analysis.** This figure displays the age-standardized rates of incidence, prevalence, and DALYs for inguinal, femoral, and abdominal hernias among older adults globally from 1990 to 2021. From top to bottom: ASDR for both sexes **(A)**, males **(D)**, and females **(G)**. ASDR for both sexes **(B)**, males **(E)**, and females **(H)**. ASDR for both sexes **(C)**, males **(F)**, and females **(I)**. ASIR: Age-Standardized Incidence Rate; ASPR: Age-Standardized Prevalence Rate; DALYs: Disability-Adjusted Life Years; ASDR: Age-Standardized Disability-Adjusted Life Year Rate; APC: Annual Percentage Change; AAPC: Average Annual Percentage Change; SDI: Sociodemographic Index.

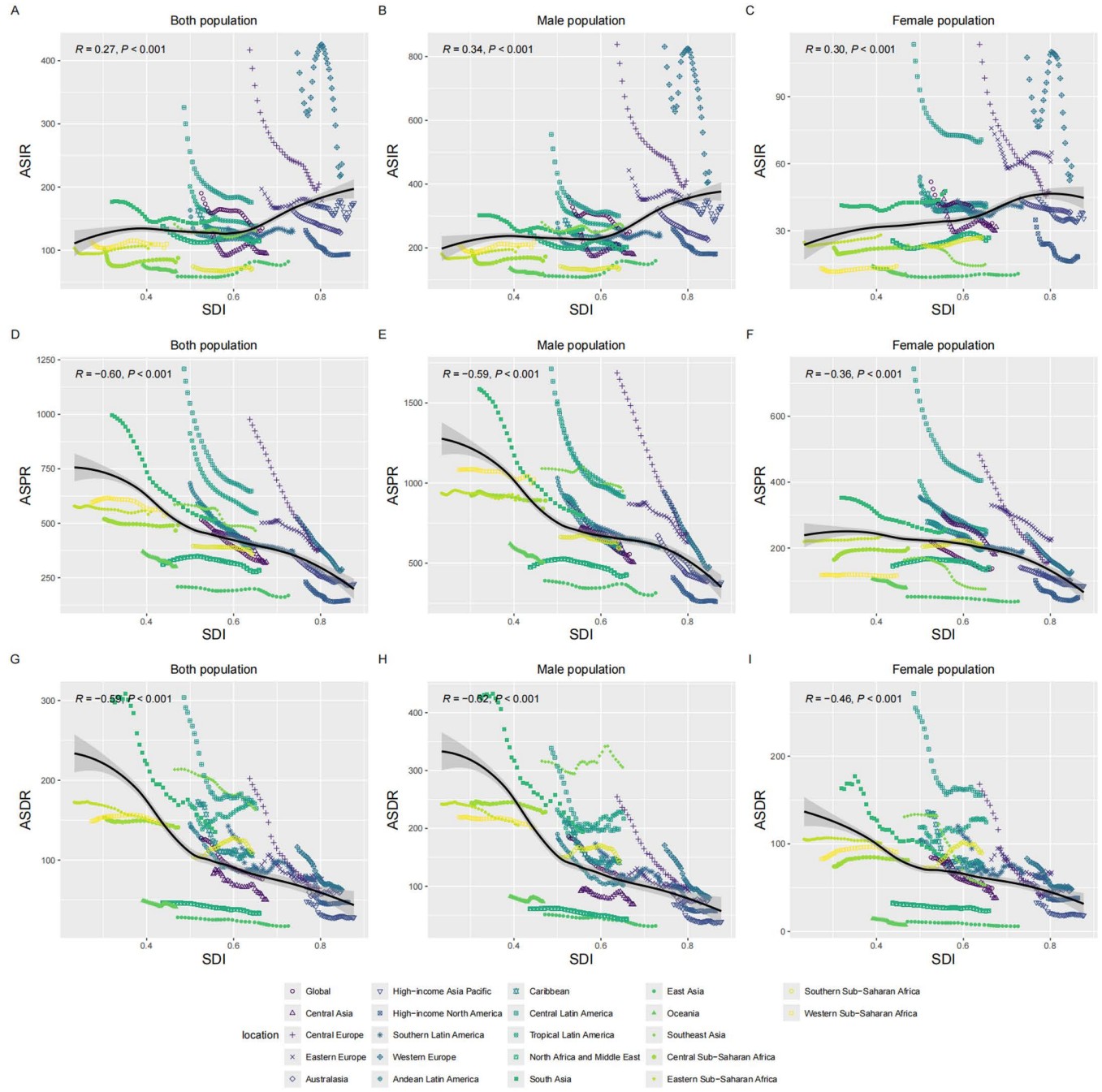

**Fig 3. Burden trends of inguinal, femoral, and abdominal hernias among older adults across regions stratified by socio-demographic index.**
This figure displays the correlation of regions to SDI and the age-standardized incidence, prevalence, and DALYs of inguinal, femoral, and abdominal hernia among older adults. From top to bottom: ASIR for both sexes **(A)**, males **(B)**, and females **(C)**. ASPR for both sexes **(D)**, males **(E)**, and females **(F)**. ASDR for both sexes **(G)**, males **(H)**, and females **(I)**. ASIR: Age-Standardized Incidence Rate; ASPR: Age-Standardized Prevalence Rate; DALYs: Disability-Adjusted Life Years; ASDR: Age-Standardized Disability-Adjusted Life Year Rate; SDI: Sociodemographic Index.

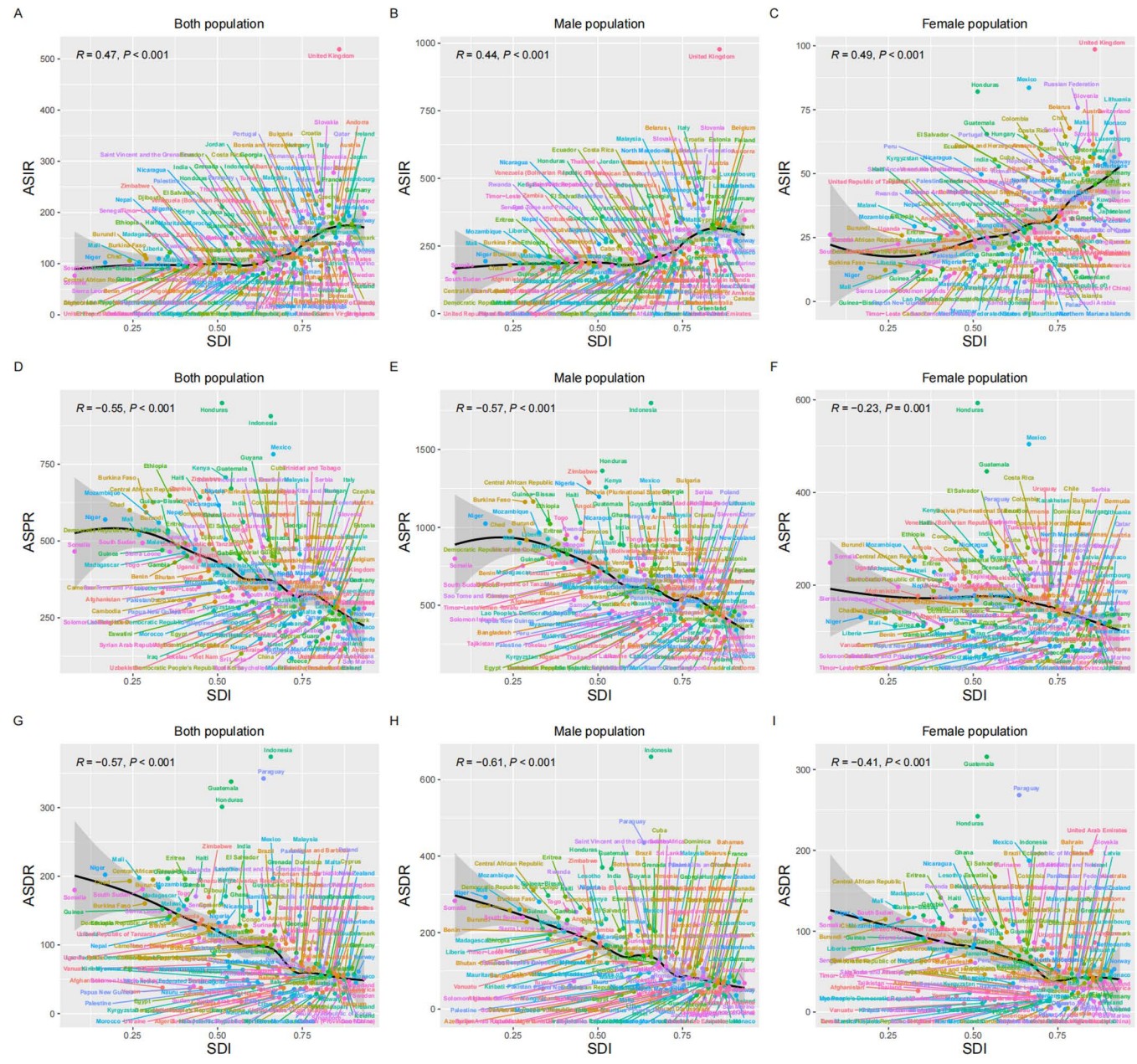

**Fig 4. Burden trends of inguinal, femoral, and abdominal hernias among older adults in countries stratified by socio-demographic index.**
This image displays the correlation of countries to SDI and the age-standardized incidence, prevalence, and DALYs of inguinal, femoral, and abdominal hernias among older adults. From top to bottom: ASIR for both sexes **(A)**, males **(B)**, and females **(C)**. ASPR for both sexes **(D)**, males **(E)**, and females **(F)**. ASDR for both sexes **(G)**, males **(H)**, and females **(I)**. ASIR: Age-Standardized Incidence Rate; ASPR: Age-Standardized Prevalence Rate; DALYs: Disability-Adjusted Life Years; ASDR: Age-Standardized Disability-Adjusted Life Year Rate; SDI: Sociodemographic Index.

## Burden trends by GBD region and country

The ASDR and ASPR showed a downward trend across all regions, with Central Europe, South Asia, Central Latin America, and Western Europe experiencing the most significant decrease. East Asia witnessed the most rapid increase in ASIR, in which the AAPC of ASIR was -1.08 (95% CI: -1.02 to -1.15, $P < 0.001$). Notably, an increase in the ASIR was

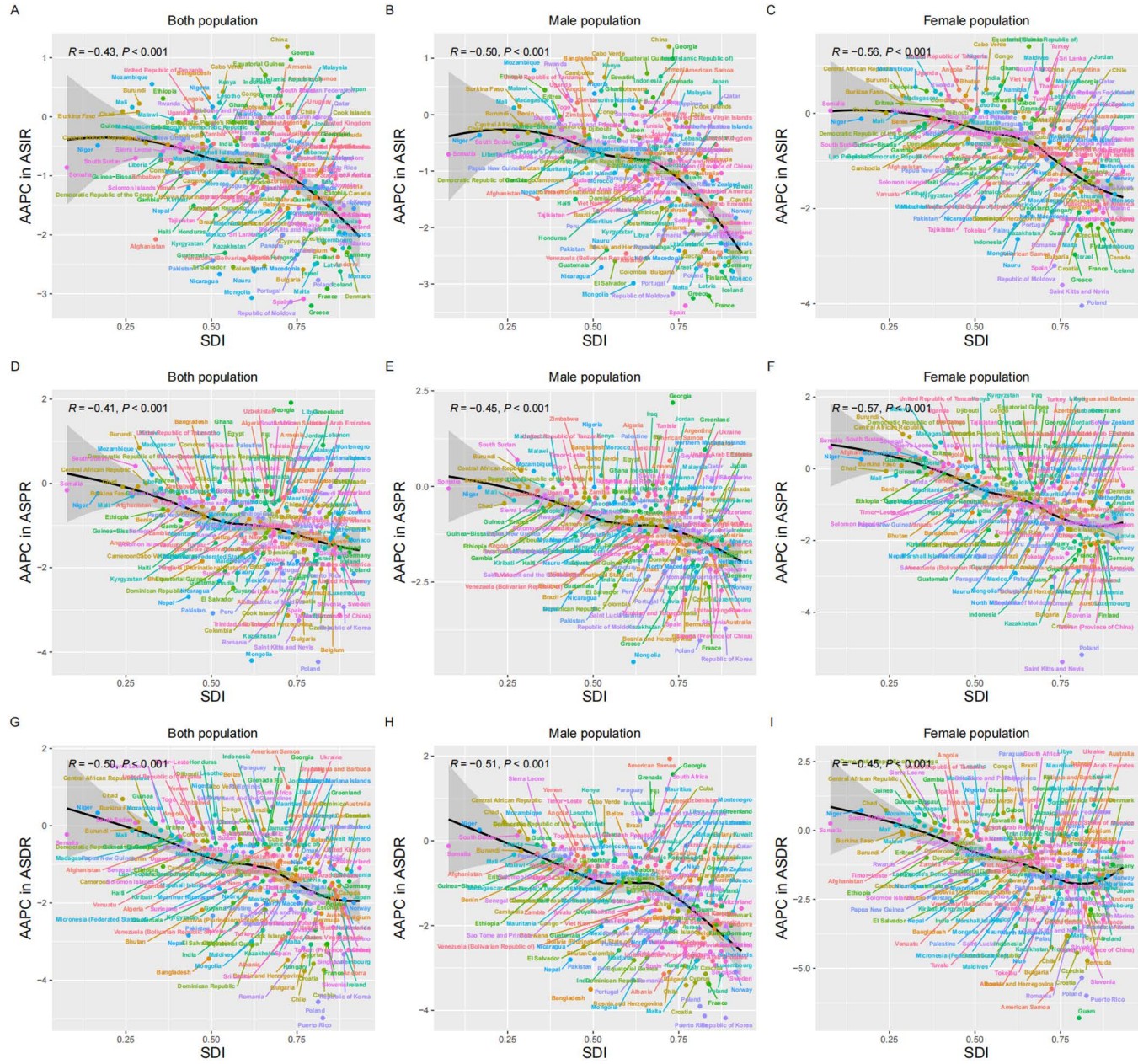

**Fig 5. Trends in AAPC of inguinal, femoral, and abdominal hernias among older adults in countries stratified by socio-demographic index.** This figure displays the correlation of countries to SDI and AAPC in the age-standardized incidence, prevalence, and DALYs of inguinal, femoral, and abdominal hernias among older adults. From top to bottom: AAPC in ASIR for both sexes **(A)**, males **(B)**, and females **(C)**. AAPC in ASPR for both sexes **(D)**, males **(E)**, and females **(F)**. AAPC in ASDR for both sexes **(G)**, males **(H)**, and females **(I)**. ASIR: Age-Standardized Incidence Rate; ASPR: Age-Standardized Prevalence Rate; DALYs: Disability-Adjusted Life Years; ASDR: Age-Standardized Disability-Adjusted Life Year Rate; SDI: Sociodemographic Index; AAPC: Average Annual Percentage Change.

observed in East Asia and European regions, including Central, Western, and Eastern Europe from 2019 to 2021. In 1990, Central Europe and Western Europe had the highest ASIR, while Central Latin America had the highest ASPR and ASDR. From 1990 to 2021, the ASIR, ASPR, and ASDR consistently decreased. However, by 2021, Central Europe and

Western Europe reported the highest ASIR, while Central Latin America kept the highest ASPR and ASDR. From 1990 to 2021, Western Europe and regions in Asia experienced the highest incident cases, prevalence, and DALYs, indicating the greatest burden (Table 1).

At national level, the overall trend of hernia among older adults was downward from 1990 to 2021. However, there was a significant increase in ASDR and ASPR in 10 countries and the ASIR in 15 countries, in which China had the highest increase in ASIR, followed by Georgia. Georgia also experienced the highest rise in ASPR and American Samoa reported the highest increase in ASDR. Additionally, there was a significant increase in ASIR and ASPR each in 70 and 72 countries from 2019 to 2021. The increase in the United Kingdom (APC in ASIR = 10.73, 95% CI: 6.23 to 15.42, $P < 0.001$) and Bangladesh (APC in ASPR = 7.79, 95% CI: 5.73 to 9.90, $P < 0.001$) was particularly notable. The United Kingdom had the highest ASIR in 1990 and 2021, and Honduras had the highest ASPR. The highest ASDR was in Guatemala in 1990, then shifted to Indonesia in 2021. In 1990 and 2021, India consistently reported the highest incident cases, prevalence, and DALYs, indicating the greatest burden (S1 Table in S1 File).

### Health inequality

Remarkable absolute and relative SDI-related inequalities in the burden of hernia among older adults were detected. In ASPR and ASDR the slope index and the concentration index have significantly increased from 1990 to 2021. This indicates that during this period, the absolute inequality in ASPR and ASDR burden among older adults in the highest and the lowest SDI countries has markedly increased, while the relative inequality was common in poorer countries. Conversely, in ASIR the slope index and the concentration index exhibited significant decrease over the same period, indicating a notable reduction in absolute inequality, and relative inequality mainly in affluent countries tending to improvements (Fig 6).

### Predictive analysis

From 1990 to 2035, the BAPC data for inguinal, femoral, and abdominal hernia among older adults showed distinct trends in ASIR, ASPR, and ASDR. The overall ASIR and ASPR initially declined significantly, followed by a steady increase since 2019. The trend for older males in ASIR and ASPR largely mirrored those of the overall population. Similarly, older females experienced stable declines in ASIR and ASPR in the early period, followed by a gradual increase since 2020. The overall ASDR displayed a downward trend in both older males and females, although the decline was slower among older females. Regardless of gender, the incident cases, prevalence, and DALYs are all expected to go upward, especially among older males (S2 Table in S1 File and Figs 7 and 8).

## Discussion

This study leverages the Global Burden of Disease 2021 database to survey the inguinal, femoral, and abdominal hernia burden among older adults. It specifically highlights the health disparities from global socio-economic inequalities and presents the substantial trends in the incidence, prevalence, and DALYs linked to these hernias from 1990 to 2021. Globally, the ASIR, ASPR and ASDR significantly declined over the study period. However, the total number of cases had increased significantly, which is consistent with previous research findings [7]. Additionally, the burden of hernia was heavier among older males than females, indicating that males suffer more risk of hernias [24,25]. It is noteworthy that ASIR and ASPR for inguinal, femoral, and abdominal hernias in older adults exhibited an upward trend after 2019. In the study of hernia burden across SDI regions, all five SDI regions showed a notable descending trend from 1990 to 2021. The most pronounced declines in ASIR, ASPR and ASDR were in high SDI regions, with ASDR showing an AAPC of -2.28, ASIR an AAPC of -1.51, and ASPR an AAPC of -2.00. Meanwhile, the absolute inequality in the ASPR and ASDR burden among older adults between the highest and lowest SDI countries significantly increased, and the relative inequality increasingly concentrated in poorer countries.

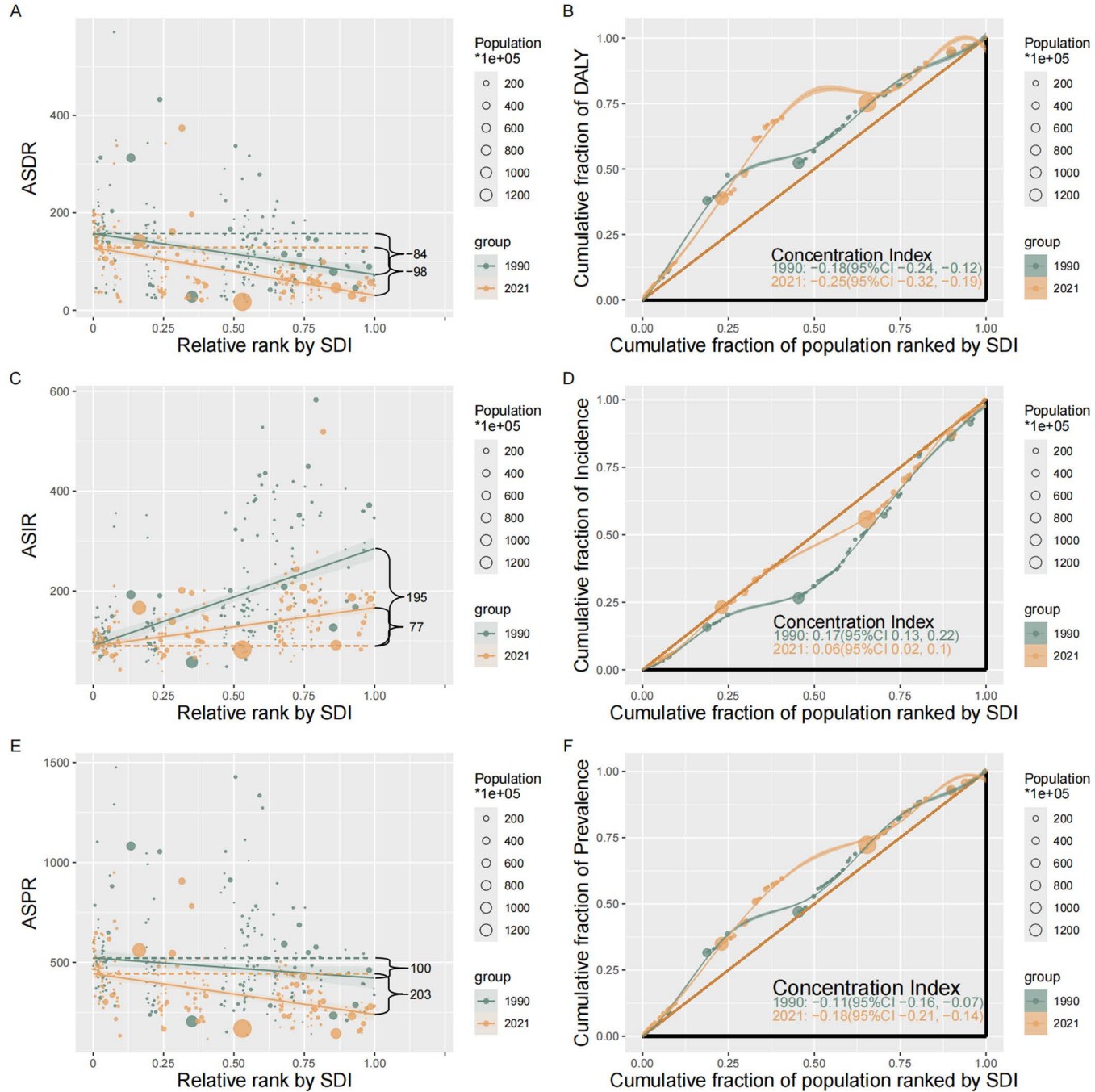

**Fig 6. Absolute healthy inequality (A, C, E) and relative healthy inequality (B, D, F) for age-standardized DALYs, incidence, and prevalence rates of inguinal, femoral, and abdominal hernias among older adults on global scale in 1990 and 2021.** The slope index of inequality, shown as the slope of the regression line, represents the absolute difference in inguinal, femoral, and abdominal hernia burden among older adults between countries or territories with the highest and lowest SDI. The concentration index, calculated as twice the area between the 45° diagonal line and the Lorenz curve, represents the relative extent to which the inguinal, femoral, and abdominal hernias burden among older adults concentrated among the poor (negative value) or the rich countries (positive value). ASIR: Age-Standardized Incidence Rate; ASPR: Age-Standardized Prevalence Rate; DALYs: Disability-Adjusted Life Years; ASDR: Age-Standardized Disability-Adjusted Life Year Rate; SDI: Sociodemographic Index.

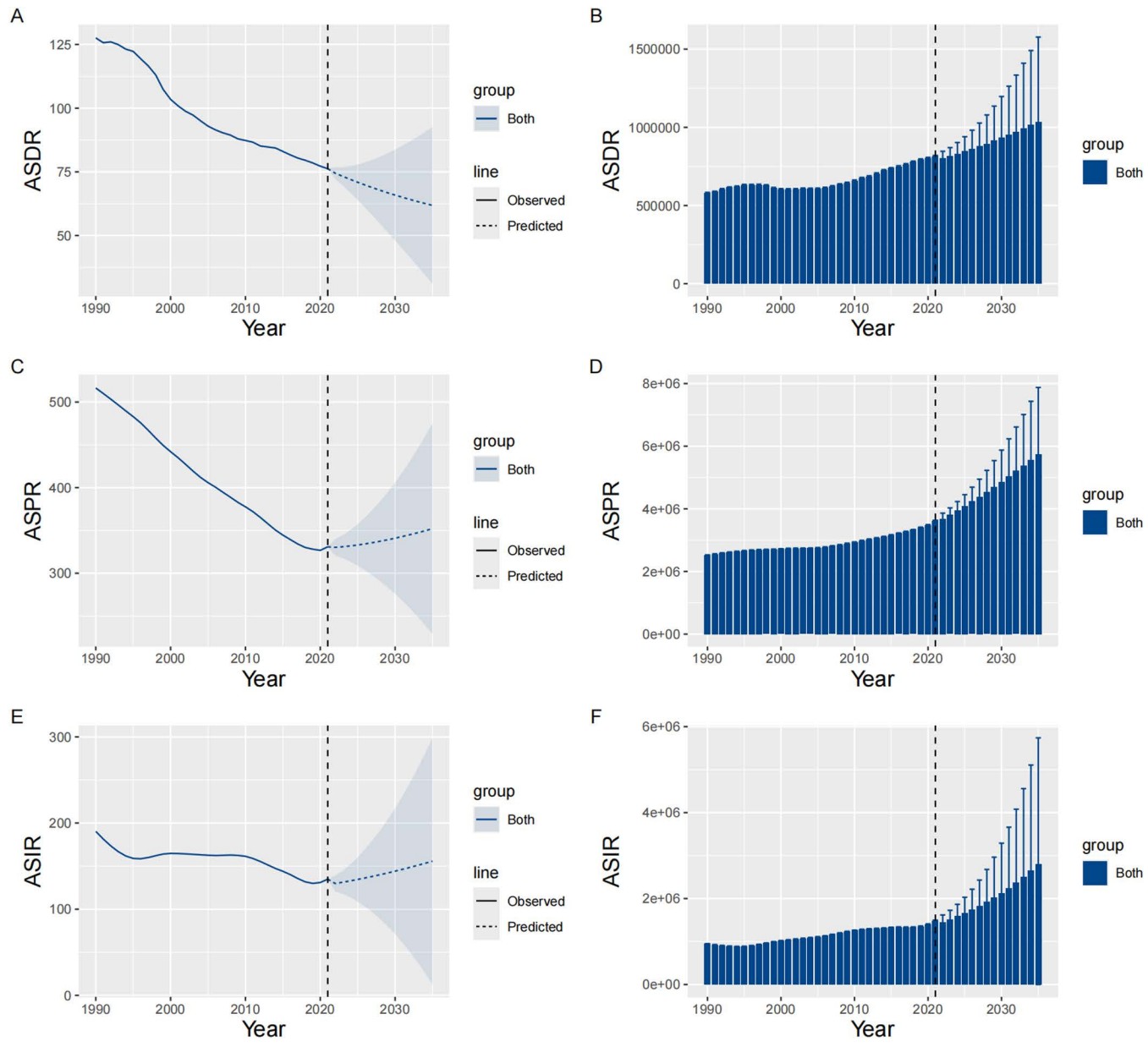

**Fig 7. Projections of inguinal, femoral, and abdominal hernia burden among older adults on global scale from 1990 to 2035.** This figure presents the observed and projected development of inguinal, femoral, and abdominal hernia burden among older adults globally from 1990 to 2035. The projections include ASR of incidence, prevalence, and DALYs, as well as the number of incident cases, prevalent cases, and DALYs sorted by gender. A: ASDR. B: Number of DALYs. C: ASPR. D: Number of prevalence cases. E: ASIR. F: Number of incidence cases. ASIR: Age-Standardized Incidence Rate; ASPR: Age-Standardized Prevalence Rate; DALYs: Disability-Adjusted Life Years; ASDR: Age-Standardized Disability-Adjusted Life Year Rate.

Our study showed that the ASIR, ASPR and ASDR for hernias in older adults decreased in 2021 compared to 1990, with AAPC values of -1.12, -1.44, and -1.64, respectively. These negative AAPC values indicate sustained, long-term reductions in both the burden and progression of hernias globally. The decrease is subject to several factors. The advancements in medical technology during the period have enabled early and accurate diagnosis, particularly through imaging techniques such as ultrasound and CT scanning [5,26]. Early diagnosis and relatively prompt treatment effectively

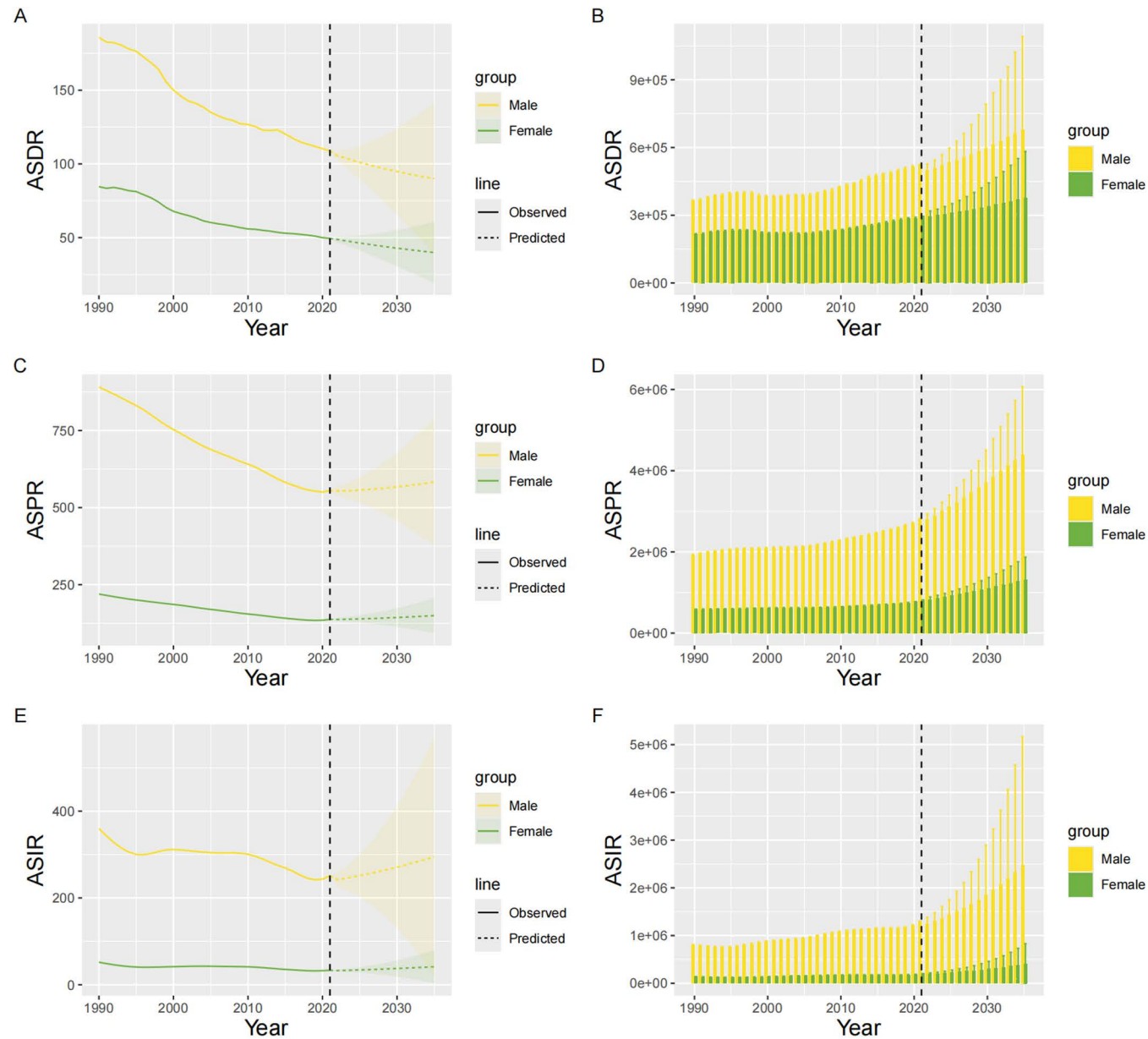

**Fig 8. Projections of inguinal, femoral, and abdominal hernias burden in global older adults population (male and female) from 1990 to 2035.**
This figure presents the observed and projected development of inguinal, femoral, and abdominal hernia burden among older adults globally from 1990 to 2035. The projections include ASR of incidence, prevalence, and DALYs as well as the number of incident cases, prevalent cases, and DALYs, sorted by gender. A: ASDR. B: Number of DALYs. C: ASPR. D: Number of prevalence cases. E: ASIR. F: Number of incidence cases. ASIR: Age-Standardized Incidence Rate; ASPR: Age-Standardized Prevalence Rate; DALYs: Disability-Adjusted Life Years; ASDR: Age-Standardized Disability-Adjusted Life Year Rate.

reduce severe complications and consequently the DALYs. The HerniaSurge Group strongly recommends the use of mesh-based repair techniques to treat inguinal and femoral hernias in its international guidelines [5,6,27]. Minimally invasive procedures have also become common practice for hernia repair [28]. The application of these technologies reduces surgical risks and shortens recovery time, thereby improving patient outcomes. The release of the WHO "World Report on

Ageing and Health" has emphasized older adult health in several developed countries [13]. Governments in high SDI have introduced a series of policies to promote active health education and disease screening, and public health awareness has gradually improved, encouraging the avoidance of strenuous physical labor and timely treatment of chronic conditions such as persistent cough, constipation, and benign prostatic hyperplasia [29–31]. These measures have played a significant role in reducing the disease burden of hernias among older adults.

Our study found that, although the ASIR has been steadily declining over time, particularly between 1990 and 2019, the absolute number of hernia cases among older adults has continued to rise. This apparent contradiction is primarily driven by global population aging. Between 1990 and 2019, the world population grew from 5.32 to 7.70 billion [12], while the number of people aged 60 years and older nearly doubled, from 538 million to 1.05 billion. Aging is a well-established risk factor for hernia development [32,33], as age-related weakening of abdominal muscles and connective tissue increases susceptibility [34]. Moreover, common comorbidities in older adults, such as chronic cough, constipation, and prostatic hypertrophy, can raise intra-abdominal pressure, further elevating hernia risk. Physiological decline tends to accelerate after age 60, making this population particularly vulnerable [35]. Given projections that the global elderly population will increase by 20% by 2050, the overall burden of hernias is expected to continue rising. Without timely health system responses, this trend may place increasing strain on surgical capacity and elderly care services worldwide.

The post-COVID-19 data help to understand the hernia burden among older adults. Compared to GBD2019, the GBD2021 reflects the impact of the pandemic on healthcare systems and provides a more comprehensive picture of global health [12,36]. Surgical delays and reduced medical services during COVID-19 lockdowns ultimately led to the progression of hernias and an increase in ASIR (APC = 1.87, 95% CI: 0.96 to 2.79, $P < 0.001$) and ASPR (APC = 0.49, 95% CI: 0.18 to 0.80, $P = 0.004$) [37,38]. The pandemic also disrupted the management of chronic diseases, exacerbating risk factors associated with hernias and promoting their progression [39]. COVID-19 may have a profound impact on the hernia burden among older adults, making the study of COVID-19 data crucial for predicting future hernia burdens and formulating prevention and treatment policies. It is suggested for future research to comprehensively examine the impact of COVID-19 on human health to enable governments to formulate more effective response policies.

This study found that older men bear a more severe burden of hernia compared to women, which is consistent with current research findings [25,32]. The inguinal canal in men is wider and shorter than in women, and as the testes descend through the inguinal canal into the scrotum, this process may lead to a congenital weakness of the canal walls, making inguinal hernias more prevalent in men [40,41]. Men are also more likely to engage in occupations that involve heavy physical labor and excessive load-bearing, increasing intra-abdominal pressure. With advancing age, this can result in a further weakening of the abdominal wall, causing or exacerbating hernia formation. Additionally, some studies suggest that male hormones, such as testosterone, may influence connective tissues, altering their structure and function, thereby affecting the strength and elasticity of the abdominal wall [42]. Governments should formulate and implement national hernia prevention strategies, focusing on the high-risk group of older men, by encouraging regular health check-ups and hernia screenings and increasing financial investment. Some studies suggest that women account for approximately 9% of inguinal hernia repair surgeries, but they face higher risks of postoperative recurrence and complications [43]. Moreover, the incidence of femoral hernias is significantly higher in women than in men, with a ratio of approximately 10:1 [44]. Although femoral hernias are relatively rare, accounting for only 2–4% of all inguinal hernias, they are more likely to result in serious complications such as strangulation and incarceration, increasing the disease burden [45]. Therefore, greater investment in the treatment of hernias in older women is also warranted.

In terms of hernia burden across different SDI regions, the high-SDI regions in 2021 saw the most significant declines in ASIR, ASPR, and ASDR. These regions likely benefited from the widespread adoption of mesh repair and improvements in surgical techniques. Mesh repair accounts for 97.6% of inguinal hernia repairs in the UK, relatively to 50% in low- and middle-income countries (LMICs), with the lowest adoption in Africa [46]. However, ASPR and ASDR in low-SDI and low-middle-SDI regions remained the highest in both 1990 and 2021. This is consistent with the findings from health inequality

analyses, indicating that the hernia burden among older adults in these areas remains heavy. This reflects deeply rooted structural disparities within health systems. Although the global burden of hernias has shown a declining trend, low-SDI regions continue to face substantial challenges in the timely diagnosis and surgical management of hernias among older adults. A key contributing factor is the limited access to safe, affordable, and timely surgical care. According to the Lancet Commission on Global Surgery, more than 5 billion people worldwide lack access to essential surgical services, with the majority residing in low-SDI settings [47]. In these regions, surgical infrastructure remains underdeveloped, and there is a critical shortage of trained surgeons, anesthesiologists, and perioperative personnel. Additionally, in these resource-limited countries, healthcare priorities are often focused on reducing mortality from infectious diseases including tuberculosis, HIV, and malaria, as well as treating non-communicable diseases like cancer and cardiovascular conditions [48]. Diseases like inguinal hernia among older adults are often of less priority in spite of its high portion of the overall disease burden and serious health impacts [49]. Looking ahead, global efforts are urgently necessary to improve healthcare infrastructure to enable surgical opportunities for older adults in low-SDI and low-middle-SDI regions, thereby reducing the hernia burden on global scale.

At region level, the consistently high ASIR in Central and Western Europe may reflect better case detection and diagnostic capacity, especially among older adults undergoing routine health evaluations. In contrast, the persistently high ASPR and ASDR in Central Latin America likely indicate a combination of limited access to timely surgical care and longer disease duration due to treatment delays. The recent increase in ASIR in East Asia is particularly noteworthy. This trend may be partly explained by rapid population aging, increased health service utilization, and improved case detection in countries such as China, where health system infrastructure has expanded significantly in recent decades [50]. However, this increase could also reflect shifts in health-seeking behavior or greater awareness of hernias among older individuals.

At national level, ASIR and ASPR each significantly increased in 70 and 72 countries from 2019 to 2021. Notably, the increases were particularly significant in UK and Bangladesh. This is closely related to the collapse of healthcare systems in these countries during the COVID-19 pandemic, where the postponement of elective surgeries led to a backlog of cases [51,52]. From 1990 to 2021, China experienced the highest increase in ASIR, while Central Latin America continued to have the highest ASPR and ASDR, and India consistently recorded the highest incidence, prevalence, and DALYs [1]. These countries and regions have large populations, and the medical resources are insufficient to meet the high demand for surgical care, leading to a heavy hernia burden among older adults. Strong healthcare policies are needed to enable more older individuals to access high-quality surgical care. Additionally, investing in healthcare infrastructure, training medical professionals, and deploying mobile surgical units can improve access to care and treatment outcomes in underserved areas. Global health partnerships should focus on resource allocation and technical assistance to strengthen healthcare systems in the most burdened regions.

The prolonged duration of COVID-19 has significantly impacted the disease burden. Compared to GBD 2019, the GBD 2021 dataset provides information on the post-COVID-19 period. To enable more accurate forecasting of future inguinal hernia burden among the elderly, we employed the BAPC model in our analysis. The predictive analysis from 1990 to 2035 shows that ASIR and ASPR for inguinal, femoral, and abdominal hernias initially experienced a significant decline but steadily rose from 2019. This suggests that the global challenges, particularly the COVID-19 pandemic, have reversed some of the prior progress made in the prevention and control of hernia diseases among older adults. As the population continues to age further, with a sustained rise in the incidence of chronic diseases among older adults [53], this trend of ASIR and ASPR is likely to persist in the future. The targeted preventive interventions for older population remain crucial, particularly the early detection and timely surgical treatment. The predictive analysis underscores the importance of prospective research in healthcare planning. Policymakers and healthcare providers are anticipated to address demographic changes and potential global challenges to maintain and improve hernia management outcomes.

In spite of the comprehensiveness of the GBD dataset, this study has certain limitations. The reliance on existing databases might introduce biases stemming from data collection methods and data accuracy across different regions.

In particular, low- and middle-income regions such as sub-Saharan Africa and parts of Southeast Asia may face challenges due to underdeveloped health information systems, resulting in underreporting of cases and deaths, incomplete vital registration, and inconsistent use of ICD coding by healthcare providers and institutions. Although the GBD study applies garbage code redistribution, covariate-based modeling, and rigorous data quality assessments to address these issues, residual uncertainties may still affect the accuracy of disease burden estimates in certain regions. The study's retrospective nature constrains certain causal inferences, and the observed trends might not fully account for unrecorded socioeconomic and environmental factors impacting hernia epidemiology. The GBD2021 includes data on elderly hernia cases related to COVID-19, and the impacts of COVID-19 on the healthcare system and elderly patients may both affect the accuracy of predictive analysis. Future research should focus on conducting prospective studies by incorporating new data sources to validate and extend these findings.

In summary, this study emphasizes on the critical need for targeted interventions and investments on healthcare infrastructure to address the disparities in hernia burden globally.

## Conclusions

This study showed the declining trend of the burden of inguinal, femoral, and abdominal hernias in global older population from 1990 to 2021. However, global disparities prevail across countries and regions. The burden of hernia among older adults has significantly decreased in high SDI regions, but it remains considerably higher in low SDI areas. Older males have greater hernia burden and risk relative to older females. The COVID-19 pandemic imposed a negative impact on the prevention and control of hernia diseases among older adults. There is a need for intensified monitoring of hernias as well as the improvement of healthcare infrastructure. Overall, developing robust and targeted public health policies is crucial to alleviate the burden of hernias among older adults effectively.

## Supporting information

**S1 File. S1 Table. Incidence, prevalence and DALYs of inguinal, femoral, and abdominal hernias among older adults in 1990 and 2021, and AAPC from 1990 to 2021, by countries. S2 Table. Predictions of inguinal, femoral, and abdominal hernias among older adults: age-standardized incidence, prevalence and DALYs rates and total cases to 2035 with 95% uncertainty intervals. Please refer to the additional submitted attachment for details.** (DOCX)

## Acknowledgments

We appreciate the outstanding work of the Global Burden of Diseases, Injuries, and Risk Factors Study 2021 collaborators. And we are deeply grateful to Dr. Xuanchen Liu and Dr. Yingda Song for his valuable suggestions that significantly improved this manuscript.

## Author contributions

**Conceptualization:** Jinwei Zhang.

**Data curation:** Jinwei Zhang.

**Formal analysis:** Jinwei Zhang, Jiaxuan Wang.

**Funding acquisition:** Yonghong Dong.

**Investigation:** Jinwei Zhang, Jiaxuan Wang.

**Methodology:** Xiangyu Han, Junjie Fan, Chao Huang.

**Project administration:** Jinwei Zhang, Yonghong Dong.

**Resources:** Jinwei Zhang, Yonghong Dong.

**Software:** Jinwei Zhang, Jiaxuan Wang.

**Supervision:** Yonghong Dong.

**Validation:** Xiangyu Han, Junjie Fan, Chao Huang, Yonghong Dong.

**Visualization:** Jinwei Zhang.

**Writing – original draft:** Jinwei Zhang.

**Writing – review & editing:** Xiangyu Han, Junjie Fan, Chao Huang, Yonghong Dong.

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
