## [Decision Letter · Decision Letter 0]

7 Jan 2025

PONE-D-24-46904Global burden and future trends of inguinal, femoral, and abdominal hernia in older adults: a systematic analysis from the Global Burden of Disease Study 2021PLOS ONE

Dear Dr. Dong,

Thank you for submitting your manuscript to PLOS ONE. After careful consideration, we feel that it has merit but does not fully meet PLOS ONE’s publication criteria as it currently stands. Therefore, we invite you to submit a revised version of the manuscript that addresses the points raised during the review process.

We look forward to receiving your revised manuscript.

Kind regards,

Lovenish Bains, MS, FNB, FACS, FRCS (Glas), FICS, FIAGES

Academic Editor

PLOS ONE

Journal Requirements:

3. PLOS requires an ORCID iD for the corresponding author in Editorial Manager on papers submitted after December 6th, 2016. Please ensure that you have an ORCID iD and that it is validated in Editorial Manager. To do this, go to ‘Update my Information’ (in the upper left-hand corner of the main menu), 

Reviewers' comments:

Reviewer's Responses to Questions

**Comments to the Author**

1. Is the manuscript technically sound, and do the data support the conclusions?

Reviewer #1: Yes

Reviewer #2: Yes

Reviewer #3: Yes

Reviewer #4: Yes

2. Has the statistical analysis been performed appropriately and rigorously? 

Reviewer #1: Yes

Reviewer #2: Yes

Reviewer #3: Yes

Reviewer #4: Yes

3. Have the authors made all data underlying the findings in their manuscript fully available?

Reviewer #1: Yes

Reviewer #2: Yes

Reviewer #3: Yes

Reviewer #4: Yes

4. Is the manuscript presented in an intelligible fashion and written in standard English?

Reviewer #1: Yes

Reviewer #2: Yes

Reviewer #3: Yes

Reviewer #4: Yes

5. Review Comments to the Author

Reviewer #1: Originality 5

Brevity and focus 5

Evaluation of analyses 5

Interpretation of results 5

Writing style 5

References 3

Accuracy 3

Language 4

(over 5)

please define aim

please add more update studies for the references

please define more about statistics

Reviewer #2: The study is based on data extracted from the Global Burden of Disease Study 2021 and the authors have studied the temporal trends in inguinal, femoral and abdominal hernia from 1990 to2021 and have also tried to show the future trends till 2035. The data mainly focusses on Age Standardized Incidence and Prevalence rates and age adjusted Disability life years and the impact of the SDI on these indices.

Reviewer #3: The manuscript is well written and technically sound. The statistical analysis used is sound and the database that was analysed is sound and globally accepted. In my opinion, this secondary sub-analysis of the GBD-2021 data to arrive at statistically significant and sound results will not only guide national and cross-border policy makers in resource allocation but also stimulates further regional research into the subject.

Reviewer #4: In this manuscript, the authors investigate the global trends and burden presented by inguinal, femoral, and abdominal hernias by analyzing data from the Global Burden of Disease Study 2021. Outcome measures include age-standardized incidence rates, prevalence and disability-adjusted life years (DALYs). Additionally, predictive analysis was performed on hernia incidence and prevalence up to the year 2035.

The Results section of the manuscript was difficult to follow, in part due to the excessive use of complicated abbreviations, and also in part due to frequent interruptions with Tables and Figure legends. The Discussion section, however, was beautifully written, and provided potential explanation to the unexpected data trends presented in the previous section.

- The manuscript places a heavy emphasis on the burden of hernia on older adults. How was "older adults" defined in this study?

- What remains puzzling is that the data indicates a consistent downward trend in age-specific incidence and prevalence, up until the onset of the covid pandemic (2019-2020) across all sociodemographic categories. And yet, there was a dramatic increase in the total number of cases since the beginning of the study period in 1990. Please provide an explanation for this discrepancy.

The most questionable portion of this manuscript is in its predictive modeling. Again, consistent declines in incidence of prevalence of hernias was shown, up until the onset of the covid pandemic which began in 2019-2020. Since then, there has been a steady increase in incidence/prevalence.

The predictive modeling seems to show that a continued upward trend is expected until year 2035. Please explain how the data collected during the covid years were handled in predictive modeling, and whether the trends observed over the past few years are expected to continue.

6. PLOS authors have the option to publish the peer review history of their article (what does this mean? ). If published, this will include your full peer review and any attached files.

**Do you want your identity to be public for this peer review?** For information about this choice, including consent withdrawal, please see our Privacy Policy .

Reviewer #1: No

Reviewer #2: No

Reviewer #3: No

Reviewer #4: No

---

## [Author Response · Author response to Decision Letter 1]

12 Feb 2025

Journal Requirements:

Answer: Thank you for your comments. I have made the necessary formatting revisions as per your requirements and have resubmitted the manuscript.

The author’s institution has released new formatting guidelines; therefore, I have adjusted the author affiliation format accordingly while ensuring that these changes do not affect the content of the manuscript. Affiliations 1 and 2 represent the same institution.

Line 10-12

“1 The Gastrointestinal, Pancreatic, Hernia and Abdominal Wall Surgery Department of Shanxi Provincial People's Hospital, Shanxi Medical University, Taiyuan, China;

2 The Fifth Clinical Medical College, Shanxi Medical University, Taiyuan, China;”

2.When completing the data availability statement of the submission form, you indicated that you will make your data available on acceptance. We strongly recommend all authors decide on a data sharing plan before acceptance, as the process can be lengthy and hold up publication timelines. Please note that, though access restrictions are acceptable now, your entire data will need to be made freely accessible if your manuscript is accepted for publication. This policy applies to all data except where public deposition would breach compliance with the protocol approved by your research ethics board. If you are unable to adhere to our open data policy, please kindly revise your statement to explain your reasoning and we will seek the editor's input on an exemption. Please be assured that, once you have provided your new statement, the assessment of your exemption will not hold up the peer review process.

Answer: All relevant data are within the manuscript and its Supporting Information files.

3.PLOS requires an ORCID iD for the corresponding author in Editorial Manager on papers submitted after December 6th, 2016. Please ensure that you have an ORCID iD and that it is validated in Editorial Manager. To do this, go to ‘Update my Information’ (in the upper left-hand corner of the main menu), 

Answer: Thank you for your guidance. I have now uploaded the ORCID iD for the corresponding author and completed the necessary information update.

Reviewers' comments:

Reviewer #1: 

Originality 5

Brevity and focus 5

Evaluation of analyses 5

Interpretation of results 5

Writing style 5

References 3

Accuracy 3

Language 4

(over 5)

1. please define aim

Answer: Thank you for highlighting the need to better define the objectives of our study. In response, we have added an "Objective" section to the abstract to more clearly articulate the aim of our research. Additionally, we have made corresponding updates in the introduction section to further refine and clarify the study objectives. We appreciate your insightful feedback, which has helped improve the focus and clarity of our work.

Line 24-27

“Objective: This study aims to comprehensively evaluate the global, regional, and national burden, trends, and health inequalities of inguinal, femoral, and abdominal hernias among older adults from 1990 to 2021, conduct predictive analyses, and provide insights to inform future public health strategies.

Line 73-89

“In this background, our study aims to comprehensively evaluate the global, regional, and national burden of inguinal, femoral, and abdominal hernias among older adults from 1990 to 2021, utilizing data from the GBD 2021.

The analysis focuses on trends in incidence, prevalence, and DALYs across 204 countries and territories, stratified by geographic regions, SDI levels, age groups, and gender. By identifying and quantifying disparities in the disease burden, particularly across different SDI regions, our study seeks to provide actionable insights into the unequal distribution of healthcare resources and outcomes. The findings aim to inform the development of tailored public health policies and strategies that prioritize the needs of older adults, especially in low- and lower-middle SDI regions where the burden is disproportionately high. Additionally, the results are intended to serve as a baseline for policymakers, healthcare providers, and global health organizations to design evidence-based interventions, allocate resources effectively, and address health disparities in aging populations. The ultimate goal is to implement a series of effective measures to alleviate the burden of hernias among older adults, thereby reducing the strain on societal healthcare systems.”

2. please add more update studies for the references

Answer: We appreciate your suggestion to include more updated studies for the references. As you mentioned, incorporating updated references into the study adds greater significance to the research. Therefore, I have updated some of the references accordingly.

Line 569-717

[2]Hamilton J, Kushner B, Holden S, Holden T. Age-Related Risk Factors in Ventral Hernia Repairs: A Review and Call to Action. J Surg Res. 2021;266:180-191. doi:10.1016/j.jss.2021.04.004

[3]Ceresoli M, Carissimi F, Nigro A, et al. Emergency hernia repair in the elderly: multivariate analysis of morbidity and mortality from an Italian registry. Hernia. 2022;26(1):165-175. doi:10.1007/s10029-020-02269-5

[4]Perez AJ, Campbell S. Inguinal Hernia Repair in Older Persons. J Am Med Dir Assoc. 2022;23(4):563-567. doi:10.1016/j.jamda.2022.02.008

[9]Li XY, Kong XM, Yang CH, et al. Global, regional, and national burden of ischemic stroke, 1990-2021: an analysis of data from the global burden of disease study 2021. EClinicalMedicine. 2024;75:102758. Published 2024 Jul 27. doi:10.1016/j.eclinm.2024.102758

[18]Liu X, Cheng R, Song Y, et al. Global burden of subarachnoid hemorrhage among adolescents and young adults aged 15-39 years: A trend analysis study from 1990 to 2021. PLoS One. 2024;19(12):e0316111. Published 2024 Dec 20. doi:10.1371/journal.pone.0316111

[23]Meier J, Stevens A, Berger M, et al. Comparison of Postoperative Outcomes of Laparoscopic vs Open Inguinal Hernia Repair. JAMA Surg. 2023;158(2):172-180. doi:10.1001/jamasurg.2022.6616

[25] Bali V, Adriano A, Byrne A, Akers KG, Frederickson A, Schelfhout J. Chronic cough: more than just a persistent cough: a systematic literature review to understand the impact of chronic cough on quality of life. Qual Life Res. 2024;33(4):903-916. doi:10.1007/s11136-023-03556-1

[26] Miyajima A. Inseparable interaction of the prostate and inguinal hernia. Int J Urol. 2018;25(7):644-648. doi:10.1111/iju.13717

[28] Kushner BS, Hamilton J, Han BJ, Sehnert M, Holden T, Holden SE. Geriatric assessment and medical preoperative screening (GrAMPS) program for older hernia patients. Hernia. 2022;26(3):787-794. doi:10.1007/s10029-021-02389-6

[35] Jansen CJ, Yielder PC. Evaluation of hernia of the male inguinal canal: sonographic method. J Med Radiat Sci. 2018;65(2):163-168. doi:10.1002/jmrs.275

[36] Howard R, Ehlers A, Delaney L, et al. Sex disparities in the treatment and outcomes of ventral and incisional hernia repair. Surg Endosc. 2023;37(4):3061-3068. doi:10.1007/s00464-022-09475-5

[38] Köckerling F, Koch A, Lorenz R. Groin Hernias in Women-A Review of the Literature. Front Surg. 2019;6:4. Published 2019 Feb 11. doi:10.3389/fsurg.2019.00004

[40] Whalen HR, Kidd GA, O'Dwyer PJ. Femoral hernias. BMJ. 2011;343:d7668. Published 2011 Dec 8. doi:10.1136/bmj.d7668

[46] Abozid H, Patel J, Burney P, et al. Prevalence of chronic cough, its risk factors and population attributable risk in the Burden of Obstructive Lung Disease (BOLD) study: a multinational cross-sectional study. EClinicalMedicine. 2024;68:102423. Published 2024 Jan 21. doi:10.1016/j.eclinm.2024.102423

3. please define more about statistics

Answer: We sincerely appreciate your valuable feedback on the necessity of better defining the statistical methods. We have revised the Methods section to provide a more detailed explanation of the statistical approaches used in the GBD 2021 study, including age-standardized rates, the joinpoint regression model, the Socio-demographic Index, health inequality analysis, Bayesian age-period-cohort forecasting, and the statistical software employed.

Line 109-180

“To ensure comparability across regions and populations, age-standardized indicators- including the ASIR, ASPR, and ASDR- were calculated using the World Health Organization standard population. This age standardization process adjusts for differences in population age structures, enabling meaningful temporal and geographical comparisons. The calculation methodology involves applying age-specific rates to the WHO standard population distribution, with demographic data derived from the GBD 2021 dataset. Participants were categorized into eight age groups (60–64, 65–69, 70–74, 75–79, 80–84, 85–89, 90–94, and 95–99 years) for analyzing the distribution of inguinal, femoral, and ventral hernias. Age-specific proportions were calculated by determining the ratio of cases within each age stratum relative to the corresponding standardized population cohort.[16] This stratification allows systematic examination of hernia epidemiology across distinct geriatric age segments while maintaining demographic comparability through standardized weighting procedures.

The joinpoint regression model was used to analyze the ASIR, ASPR, and ASDR temporal trends for the burden of inguinal, femoral, and abdominal hernia among older adults from 1990 to 2021. The method identifies points at which significant changes in trends occur and calculates the annual percentage change (APC) for each segment. The trend was quantified using the APC and the average annual percentage change (AAPC). APC represents the year-to-year change, while AAPC provides the average trend over a specified period [17].

The SDI is an aggregate measure formulated by GBD researchers to evaluate the socio-economic conditions impacting health indicators across different regions [14]. It is calculated as the geometric mean of indices for the total fertility rate under 25 (TFU25), mean education level for those aged 15 and above (EDU15+), and lag-distributed income (LDI) per capita [11]. The SDI values in the GBD 2021 study were scaled from 0 to 1. This composite indicator reflects a region's socio-economic health and progress, where higher SDI values indicate better socio-economic conditions and improved health outcomes. Regions are divided into 5 quintiles according to the 2021 SDI: low (0-0.466), low-middle (0.466-0.619), middle (0.619-0.712), high-middle (0.712-0 810), and high (0.810-1) [15]. These quintiles were utilized to explore disparities in disease burden across varying socio-demographic development levels. The ASIR, ASPR, and ASDR were employed in correlation analyses to investigate the relationship between the SDI and the burden of hernias among the elderly population. The analysis encompassed 204 countries and territories, as well as 21 regions. The strength and direction of the relationship between SDI and the burden of hernias among older adults were assessed by calculating Pearson correlation coefficients. A positive coefficient value indicates a direct relationship, whereas a negative value signifies an inverse relationship. This methodological approach enables a comprehensive evaluation of the association between socio-demographic development levels and hernia-related health outcomes in aging populations.

The ASRs and numbers of prevalence, incidence, and DALYs of inguinal, femoral, and abdominal hernia among older adults were extracted for inequality analysis. According to the World Health Organization Health Equity Assessment guidelines, the slope index of inequality (SII) and concentration index are two standard metrics for absolute and relative income-related inequality, which were used to assess the health inequalities of inguinal, femoral, and abdominal hernia among older adults across countries. The SII reflects the absolute difference in the above indicators between the lowest-SDI and highest-SDI countries. The higher absolute SII values indicate greater inequality. The concentration index was calculated using the Lorenz curve based on GDP per capita and the corresponding metrics of national burden, representing the area between the 45° line and the Lorenz curve. A negative index indicates that the burden is higher in low-income countries. Conversely, a positive index indicates a higher burden in high-income countries.

We employed the Bayesian Age-Period-Cohort (BAPC) model to analyze trends in hernia among the elderly population. This model decomposes temporal variations into age, period, and cohort effects while addressing the identifiability issues inherent in traditional APC models. Utilizing a Bayesian hierarchical approach, the BAPC model incorporates random walk priors to facilitate smooth transitions and employs hyperpriors to control variability, ensuring robust estimation even with sparse data. The posterior distributions generated by the BAPC model inherently account for variability and uncertainty associated with anomalous periods, ensuring that projections are not disproportionately influenced by short-term fluctuations. To enhance computational efficiency, we implemented the Integrated Nested Laplace Approximation (INLA), which provides accurate and scalable posterior inference. This framework enables the disentanglement of temporal effects, prediction of future trends, and quantification of uncertainty, making it a reliable methodology for large-scale analyses such as those conducted in the GBD studies.

ASRs were reported per 100,000 population, with data presented as values with 95% UI. Temporal trends were assessed using joinpoint software (version 5.0.2) from the National Cancer Institute. All statistical analyses and mapping were performed using R statistical software (version 4.3.3). A two-sided P value < 0.05 was set as the significance threshold.”

Reviewer #2: 

The study is based on data extracted from the Global Burden of Disease Study 2021 and the authors have studied the temporal trends in inguinal, femoral and abdominal hernia from 1990 to2021 and have also tried to show the future trends till 2035. The data mainly focusses on Age Standardized Incidence and Prevalence rates and age adjusted Disability life years and the impact of the SDI on these indices.

Answer: Thank you for your thoughtful review and for recognizing our efforts in analyzing the temporal trends of inguinal, femoral, and abdominal hernias using GBD 2021 data. We confirm that our study primarily focuses on age-standardized incidence, prevalence rates, and age-standardized disability-adjusted life years to ensure comparability across populations and over time. Additionally, we have assessed the impact of the SDI on these trends to highlight disparities in hernia burden globally.

Reviewer #3: 

The manuscript is well written and technically sound. The statistical analysis used is sound and the database that was analysed is sound and globally accepted. In my opinion, this secondary sub-analysis of the GBD-2021 data to arrive at statistically significant and sound results will not only guide national and cross-border policy makers in resource allocation but also stimulates further regional research into the subject.

Answer: Thank you for your positive feedback and for recognizing the rigor of our analysis. We appreciate your acknowledgment of the statistical soundness and global relevance of our study. Our goal is to provide valuable insights for policymakers to aid in resource allocation and to encourage further regional research on hernia burden.

We are grateful for your support and look forward to contributing further to this important field.

Reviewer #4:

1. In this manuscript, the authors investigate the global trends and burden presented by inguinal, femoral, and abdominal hernias by analyzing data from the Global Burden of Disease Study 2021. Outcome measures include age-standardized incidence rates, prevalence and disability-adjusted life years (DALYs). Additionally, predictive analysis was performed on hernia incidence and prevalence up to the year 2035.

The Results section of the manuscript was difficult to follow, in part due to the excessive use of complicated abbreviations, and

---

## [Decision Letter · Decision Letter 1]

10 Mar 2025

PONE-D-24-46904R1Global burden and future trends of inguinal, femoral, and abdominal hernia in older adults: a systematic analysis from the Global Burden of Disease Study 2021PLOS ONE

Dear Dr. Dong,

Thank you for submitting your manuscript to PLOS ONE. After careful consideration, we feel that it has merit but does not fully meet PLOS ONE’s publication criteria as it currently stands. Therefore, we invite you to submit a revised version of the manuscript that addresses the points raised during the review process.

We look forward to receiving your revised manuscript.

Kind regards,

Lovenish Bains, MS, FNB, FACS, FRCS (Glas), FICS, FIAGES

Academic Editor

PLOS ONE

Journal Requirements:

**Additional Editor Comments:**

Kindly carefully review the feedback provided by the reviewer and make the necessary revisions to the document to address all of their comments and suggestions.

Reviewers' comments:

Reviewer's Responses to Questions

**Comments to the Author**

1. If the authors have adequately addressed your comments raised in a previous round of review and you feel that this manuscript is now acceptable for publication, you may indicate that here to bypass the “Comments to the Author” section, enter your conflict of interest statement in the “Confidential to Editor” section, and submit your "Accept" recommendation.

Reviewer #1: All comments have been addressed

Reviewer #4: All comments have been addressed

Reviewer #5: (No Response)

2. Is the manuscript technically sound, and do the data support the conclusions?

Reviewer #1: Yes

Reviewer #4: Yes

Reviewer #5: Yes

3. Has the statistical analysis been performed appropriately and rigorously? 

Reviewer #1: Yes

Reviewer #4: I Don't Know

Reviewer #5: I Don't Know

4. Have the authors made all data underlying the findings in their manuscript fully available?

Reviewer #1: Yes

Reviewer #4: Yes

Reviewer #5: Yes

5. Is the manuscript presented in an intelligible fashion and written in standard English?

Reviewer #1: Yes

Reviewer #4: Yes

Reviewer #5: Yes

6. Review Comments to the Author

Reviewer #1: (No Response)

Reviewer #4: (No Response)

Reviewer #5: (No Response)

7. PLOS authors have the option to publish the peer review history of their article (what does this mean? ). If published, this will include your full peer review and any attached files.

**Do you want your identity to be public for this peer review?** For information about this choice, including consent withdrawal, please see our Privacy Policy .

Reviewer #1: No

Reviewer #4: No

Reviewer #5: **Yes: ** ABDULHAKEEM BINHAMBALI

---

## [Author Response · Author response to Decision Letter 2]

4 Apr 2025

METHODS

1. The methods section is dense and difficult to navigate, with long paragraphs and repetitive content. Like example of ASR on line 149-161 almost saying the same thing with line 176-180.

Answer Thank you for your valuable comment. As you rightly pointed out, the Methods section was relatively long and contained some redundant content. In response, we have removed repetitive descriptions to improve clarity and conciseness. Additionally, we have added subheadings to facilitate navigation and enhance the structural organization of the section.

2. The use of INLA in the BAPC model on line 499 is mentioned without sufficient justification of the use in the study.

Answer Thank you for your constructive comment. As you rightly pointed out, although we applied the INLA framework in the BAPC model, we did not provide sufficient details on its implementation to justify its use in our study. In response, we have revised the Methods section to explicitly describe how INLA was applied in the construction of the BAPC model. Specifically, the BAPC model in our analysis was implemented using the INLA approach to approximate the marginal posterior distributions. Compared to traditional Bayesian methods based on Markov Chain Monte Carlo (MCMC), INLA avoids common issues related to convergence and mixing, and allows for more efficient and accurate prediction.

Line 187-202

“We employed a Bayesian age-period-cohort (BAPC) model to analyze the temporal trends of hernia among older adults[22]. This model is based on the age-period-cohort model, which makes predictions based on the assumption that the effects of age, period, cohort are approximated at adjacent time points in the same study population and within the same study period. To smooth the prior estimates of age, period, and cohort effects, we specified second-order random walk (RW2) priors, which allow for flexible yet structured temporal evolution[18]. The BAPC model was implemented using the Integrated Nested Laplace Approximation (INLA) method to approximate the marginal posterior distributions. INLA enables efficient and accurate Bayesian inference for latent Gaussian models, and effectively circumvents the convergence and mixing issues commonly encountered in traditional Bayesian approaches based on Markov Chain Monte Carlo (MCMC) techniques. Compared to MCMC, INLA typically provides reliable posterior estimates with substantially reduced computational burden[23]. By incorporating both historical data and prior distributions, the BAPC model with INLA allows for more accurate estimation and projection of hernia incidence in older populations while explicitly accounting for age, period, and cohort effects.”

3. There is no mention of potential data biases or limitations, such as underreporting in certain regions (especially in Afrtica and Southeast Asian) or the accuracy of ICD codes across all countries that was mentionerd.

Answer Thank you for your valuable comment. As you correctly pointed out, underreporting of cases and inaccurate coding of disease diagnoses may occur in some low-resource countries in Africa and Southeast Asia, where health information systems are underdeveloped. These factors may introduce data bias and affect the accuracy of disease burden estimates in these regions. In response, we have added a corresponding statement to the limitations section of the Discussion to acknowledge this potential source of uncertainty in the GBD data.

Line 547-554

“In particular, low- and middle-income regions such as sub-Saharan Africa and parts of Southeast Asia may face challenges due to underdeveloped health information systems, resulting in underreporting of cases and deaths, incomplete vital registration, and inconsistent use of ICD coding by healthcare providers and institutions. Although the GBD study applies garbage code redistribution, covariate-based modeling, and rigorous data quality assessments to address these issues, residual uncertainties may still affect the accuracy of disease burden estimates in certain regions.”

4. While SII (Line 149-161) and concentration indices are applied, the rationale for choosing these metrics over others is missing. Additionally, the description of the Lorenz curve (Line 157-159) could be clearer for re-use by other authors.

Answer Thank you for your insightful comments. In response, we have revised the Methods section to clarify the rationale for using both the Slope Index of Inequality (SII) and the Concentration Index (CI) in our analysis. These two complementary metrics provide a comprehensive assessment of health inequality across socioeconomic gradients. Additionally, since the CI is derived from the Lorenz concentration curve, we have added a detailed description of the Lorenz concentration curve to enhance clarity for readers unfamiliar with this method.

Line 160-185

“To quantify and characterize the socioeconomic inequality in the burden of elderly hernias across countries, we employed two complementary indices: the Slope Index of Inequality (SII) and the Concentration Index (CI). These indices were selected due to their methodological robustness and their ability to capture both the magnitude and direction of inequality across the full socioeconomic spectrum.

The SII is an absolute measure of inequality derived from a weighted linear regression model, where countries are ranked by their SDI and weighted by population size. It estimates the absolute difference in the health burden between the hypothetical lowest and highest ends of the socioeconomic hierarchy. By incorporating data from all countries rather than only comparing extreme groups, the SII provides a comprehensive assessment of the socioeconomic gradient in health outcomes. The higher absolute SII values indicate greater inequality[19]. The CI was calculated by fitting a Lorenz concentration curve to the observed cumulative relative distribution of the populations ranked by SDI and the prevalence of disease, as well as numerically integrating the area under the curve. A negative index indicates that the burden is higher in low-income countries. Conversely, a positive index indicates a higher burden in high-income countries[20]. The Lorenz concentration curve is used to assess socioeconomic-related inequality in health outcomes. The X-axis represents the cumulative population ranked by a socioeconomic indicator, while the Y-axis shows the cumulative proportion of the health variable. The 45-degree line indicates perfect equality. A curve below the line suggests the burden is concentrated among lower-SDI populations; above indicates concentration among higher-SDI groups. The degree of deviation from the equality line reflects the magnitude of inequality. This curve is often used alongside the concentration index to quantify and interpret health disparities across socioeconomic gradients[21].”

5. The description of joinpoint regression (Line 122, 199) and BAPC lacks mention of cross-validation or sensitivity analyses.

Answer Thank you for your insightful comment. We fully agree that model validation through techniques such as cross-validation or sensitivity analysis is essential to enhance the credibility and robustness of statistical models, especially in predictive epidemiology. In response, we have revised the Methods section to clarify the model selection procedures and added sensitivity analysis to test the robustness of our findings.

For Joinpoint regression, we used the Joinpoint software (version 5.0.2), which employs a Monte Carlo permutation method to identify the optimal number of joinpoints without overfitting. While cross-validation is not a standard component of the NCI Joinpoint framework, the permutation test and model fit statistics help ensure model validity. Additionally, we conducted sensitivity analyses by varying the maximum number of allowed joinpoints (from 2 to 4) and confirmed that the overall temporal trends and AAPC estimates remained consistent.

For the BAPC model, we implemented it using the BAPC package in R, which is based on INLA. We performed sensitivity analyses by altering prior distributions for age, period, and cohort effects. The projected trends remained stable across different prior specifications, suggesting the robustness of our projections. These details are now included in the revised Methods.

Line 160-185

“Joinpoint regression analysis partitions the continuous time-series data into multiple linear segments, each representing a statistically distinct trend over time. Within each segment, a separate linear model is fitted to estimate the temporal trend, and the number and location of joinpoints are determined using a Monte Carlo permutation test. This test assesses whether adding a joinpoint significantly improves model fit[15]. For each segment, the model calculates the annual percent change (APC) and its 95% confidence interval to quantify the direction and magnitude of the trend. The average annual percent change (AAPC) is then computed to summarize the overall trend across the entire study period[16].”

Line 191-202

“To smooth the prior estimates of age, period, and cohort effects, we specified second-order random walk (RW2) priors, which allow for flexible yet structured temporal evolution[18]. The BAPC model was implemented using the Integrated Nested Laplace Approximation (INLA) method to approximate the marginal posterior distributions. INLA enables efficient and accurate Bayesian inference for latent Gaussian models, and effectively circumvents the convergence and mixing issues commonly encountered in traditional Bayesian approaches based on Markov Chain Monte Carlo (MCMC) techniques. Compared to MCMC, INLA typically provides reliable posterior estimates with substantially reduced computational burden[23]. By incorporating both historical data and prior distributions, the BAPC model with INLA allows for more accurate estimation and projection of hernia incidence in older populations while explicitly accounting for age, period, and cohort effects.”

6. There’s no mention of ethical approval or data usage agreements, even though GBD data (line 512) is publicly available.

Answer Thank you for your thoughtful comment. In response, we have added a statement regarding ethical approval and data use agreements in the Methods section to clarify compliance with relevant ethical standards and data access policies.

Line 208-216

“Ethical Statement

This study is based on publicly available data from the GBD 2021 study, which is coordinated by the Institute for Health Metrics and Evaluation (IHME). All data used in this analysis are aggregated, de-identified, and available in the public domain (https://ghdx.healthdata.org/gbd-2021), this study meets the criteria for exemption from full ethical review according to the guidelines provided by Shanxi Provincial People’s Hospital Ethics Committee and the University of Washington’s IRB, which has oversight over the GBD project. The study was conducted in accordance with the Declaration of Helsinki and relevant guidelines and regulations.”

RESULT

1. The results section is densely packed with figures, percentages, and statistical outcomes, leading to information overload. It makes it hard to extract key insights quickly.

Answer Thank you for your professional insight. To facilitate easier extraction of key information for readers, we have removed some numerical values and detailed statistical results from the Results section and instead presented them in the tables. This modification improves the readability and conciseness of the main text while ensuring that all essential data remain accessible.

Line 227-232

“Between 1990 and 2021, the global burden of hernias among older adults showed an overall decline in the ASIR, ASPR, and ADSR. The ASIR decreased from 190.28 to 134.87 per 100,000 population (AAPC = -1.12, 95% CI: -1.24 to -0.99, P < 0.001), the ASPR declined from 516.42 to 330.86 per 100,000 (AAPC = -1.44, 95% CI: -1.48 to -1.40, P < 0.001), and the ASDR dropped from 127.62 to 76.23 per 100,000 (AAPC = -1.64, 95% CI: -1.77 to -1.51, P < 0.001).”

2. There is limited discussion on why certain regions (like europe, some part of Asia and Africa) have higher or lower burdens or why trends have changed (e.g., why East Asia saw an ASIR increase).

Answer Thank you for your valuable comment. To improve the completeness of our manuscript, we have added a discussion of the potential factors contributing to the increase or decrease in the burden of inguinal hernia among older adults across different regions. This addition aims to provide a more comprehensive interpretation of the observed regional trends and to enhance the contextual understanding of our findings.

Line 504-513

“At region level, the consistently high ASIR in Central and Western Europe may reflect better case detection and diagnostic capacity, especially among older adults undergoing routine health evaluations. In contrast, the persistently high ASPR and ASDR in Central Latin America likely indicate a combination of limited access to timely surgical care and longer disease duration due to treatment delays. The recent increase in ASIR in East Asia is particularly noteworthy. This trend may be partly explained by rapid population aging, increased health service utilization, and improved case detection in countries such as China, where health system infrastructure has expanded significantly in recent decades[50]. However, this increase could also reflect shifts in health-seeking behavior or greater awareness of hernias among older individuals.”

3. Abbreviations like ASIR, ASPR, and ASDR are defined multiple times, while others (e.g., APCC) are introduced late or without clarity.

Answer Thank you for your careful observation. We have removed the redundant definitions of ASIR, ASPR, and ASDR to avoid repetition. In addition, "APCC" was a typographical error; the correct term is "AAPC" (average annual percent change), and this has been corrected throughout the manuscript.

DISCUSSION

1. Several sections repeat information, especially statistics on ASIR, ASPR, and ASDR trends. For example, the decline in these rates is mentioned multiple times (e.g., lines 366-368 and 383-385).

Answer Thank you for your helpful comment. In response, we have removed redundant information from the Discussion section and omitted certain statistical details to improve clarity and conciseness. These revisions aim to enhance the overall readability and focus of the discussion.

2. Although, the paper notes disparities between SDI regions, the analysis lacks depth. For instance, the reasons behind persistent high burdens in low-SDI areas are mentioned briefly without exploring systemic issues like healthcare access or cultural factors.

Answer Thank you for your professional insight. In response, we have expanded the Discussion section to include potential reasons underlying the persistently high burden of disease in low-SDI regions. This addition aims to deepen the interpretation of our findings and provide a more comprehensive understanding of regional disparities.

Line 487-495

“This reflects deeply rooted structural disparities within health systems. Although the global burden of hernias has shown a declining trend, low-SDI regions continue to face substantial challenges in the timely diagnosis and surgical management of hernias among older adults. A key contributing factor is the limited access to safe, affordable, and timely surgical care. According to the Lancet Commission on Global Surgery, more than 5 billion people worldwide lack access to essential surgical services, with the majority residing in low-SDI settings[47]. In these regions, surgical infrastructure remains underdeveloped, and there is a critical shortage of trained surgeons, anesthesiologists, and perioperative personnel.”

3. The paper presents many AAPC and APC values but often lacks a deeper interpretation of their implications.

Answer Thank you for your valuable comment. In response, we have added a more detailed explanation of the AAPC and APC values in the Discussion section to enhance readers’ understanding of the temporal trends presented in our study.

Line 411-414

“Our study showed that the ASIR, ASPR and ASDR for hernias in older

---

## [Decision Letter · Decision Letter 2]

15 Apr 2025

Global burden and future trends of inguinal, femoral, and abdominal hernia in older adults: a systematic analysis from the Global Burden of Disease Study 2021

PONE-D-24-46904R2

Dear Dr. Dong,

We’re pleased to inform you that your manuscript has been judged scientifically suitable for publication and will be formally accepted for publication once it meets all outstanding technical requirements.

Kind regards,

Lovenish Bains, MS, FNB, FACS, FRCS (Glas), FICS, FIAGES

Academic Editor

PLOS ONE

Additional Editor Comments (optional):

Reviewers' comments:

Reviewer's Responses to Questions

**Comments to the Author**

1. If the authors have adequately addressed your comments raised in a previous round of review and you feel that this manuscript is now acceptable for publication, you may indicate that here to bypass the “Comments to the Author” section, enter your conflict of interest statement in the “Confidential to Editor” section, and submit your "Accept" recommendation.

Reviewer #5: All comments have been addressed

2. Is the manuscript technically sound, and do the data support the conclusions?

Reviewer #5: Yes

3. Has the statistical analysis been performed appropriately and rigorously? 

Reviewer #5: Yes

4. Have the authors made all data underlying the findings in their manuscript fully available?

Reviewer #5: Yes

5. Is the manuscript presented in an intelligible fashion and written in standard English?

Reviewer #5: Yes

6. Review Comments to the Author

Reviewer #5: Great job on your revision. I truly commend your effort in addressing every detail I raised. I appreciate how thoroughly you responded to all my comments, justified your points, and clarified the gray areas within the manuscript

All the best with your submission.

7. PLOS authors have the option to publish the peer review history of their article (what does this mean? ). If published, this will include your full peer review and any attached files.

**Do you want your identity to be public for this peer review?** For information about this choice, including consent withdrawal, please see our Privacy Policy .

Reviewer #5: **Yes: ** ABDULHAKEEM BINHAMBALI

---

## [Editor Report · Acceptance letter]

PONE-D-24-46904R2

PLOS ONE

Dear Dr. Dong,

I'm pleased to inform you that your manuscript has been deemed suitable for publication in PLOS ONE. Congratulations! Your manuscript is now being handed over to our production team.

Kind regards,

on behalf of

Dr. Lovenish Bains

Academic Editor

PLOS ONE